

# Atmosphere-only GCM simulations with prescribed land surface temperatures.

Duncan Ackerley[1] and Dietmar Dommenget[1]

[1]ARC Centre of Excellence for Climate System Science, School of Earth Atmosphere and Environment, Monash University, Clayton 3800, Victoria, Australia

*Correspondence to:* Duncan Ackerley (duncan.ackerley@monash.edu)

**Abstract.** General circulation models (GCMs) are valuable tools for understanding how the global ocean-atmosphere-land surface system interacts and are routinely evaluated relative to observational datasets. Conversely, observational datasets can also be used to constrain GCMs in order to identify systematic errors in their simulated climates. One such example is to prescribe sea surface temperatures (SSTs) such that 70% of the Earth's surface temperature field is observationally constrained
(known as an Atmospheric Model Intercomparison Project, AMIP, simulation). Nevertheless, in such simulations, land surface temperatures are typically allowed to vary freely and therefore any errors that develop over the land may affect the global circulation. In this study therefore, a method for prescribing the land surface temperatures within a GCM (the Australian Community Climate and Earth System Simulator, ACCESS) is presented. Simulations with this prescribed land temperature model produce a mean climate state that is comparable to a simulation with freely varying land temperatures; for example the diurnal
cycle of tropical convection is maintained. The model is then developed further to incorporate a selection of "proof of concept" sensitivity experiments where the land surface temperatures are changed globally and regionally. The resulting changes to the global circulation in these sensitivity experiments are found to be consistent with other idealised model experiments described in the wider scientific literature. Finally, a list of other potential applications are described at the end of the study to highlight the usefulness of such a model to the scientific community.

**1 Introduction**

In order to reduce circulation errors in general circulation models (GCMs), simulations with prescribed sea surface temperatures (SST) from past observations are used (for example between 1979–2008 as part of the Atmospheric Model Intercomparison Project – AMIP: Gates, 1992; Gates et al., 1999; Taylor et al., 2012). Nevertheless, the land surface temperatures are allowed to vary freely in response to the prescribed SST fields in AMIP simulations, which means biases in the representation
of surface processes may lead to errors in the simulated atmospheric circulation. Such AMIP experiments have been developed further to include (amongst others) uniform increases of 4K to the 1979–2008 SST dataset and quadrupling carbon-dioxide concentrations with the 1979–2008 SST data (AMIP4K and AMIP4xCO$_2$, respectively Bony et al., 2011; Taylor et al., 2012); however, prescribing the land surface temperatures is not routinely done in AMIP experiments.



Previous studies that use GCMs with prescribed SSTs have shown the important role land surface temperatures play in driving the global circulation. For example, Chadwick et al. (2013b) use results from an AMIP4xCO$_2$ experiment and a GCM simulation with an increased solar constant to show that the surface warming patterns in the AMIP4xCO$_2$ cause changes in the tropical precipitation. Moreover, the meridional land surface temperature gradients over Eurasia and north Africa are

implicated in driving the Asian summer monsoon (Chou, 2003; Turner and Annamalai, 2012) and the recent recovery of Sahel rainfall (Dong and Sutton, 2015), respectively. Nevertheless, in each of the model experiments that Chadwick et al. (2013b), Chou (2003) and Dong and Sutton (2015) undertake, the land surface temperatures are allowed to vary freely in response to each of their specified boundary condition perturbations. It is then difficult to determine whether a remote (i.e. away from the region under consideration) land surface temperature response to a boundary forcing subsequently feeds back on the large scale

circulation in a way that acts to enhance or reduce the feature under consideration. By prescribing land surface temperatures in GCMs, and then perturbing them regionally and/or globally, the impact of such feedbacks can be negated somewhat. Such a GCM is described in this paper.

The aims of this study are to:

1. Document the method and code changes that are applied to a GCM in order to prescribe the land surface temperatures.

2. Show that the fully prescribed simulation produces a mean climate state that is comparable with a simulation that uses prescribed sea surface temperatures but freely evolving land surface temperatures.

3. Document the results of a series of sensitivity experiments where the land surface temperatures are perturbed.

4. Show that the atmospheric responses in those perturbation experiments are physically plausible and agree with the results of other studies in the literature.

5. Overall, provide a "proof of concept" by attaining the aims above and show that GCM simulations with prescribed land surface temperature are realistic and have many potential applications.

It should be noted that the experiments in this paper are designed to be sensitivity tests to identify whether the model atmosphere responds in a physically realistic way to the imposed land surface temperature field. The experiments are not designed to answer specific questions about the processes at work but to highlight the types of experiment that can be run with such a model setup.

The model and methods used in this study are given in Section 2, which includes descriptions of the source code changes, the development of the land temperature dataset and the experiments undertaken. An overview of the salient results for the global and regional surface air temperature, precipitation (including the diurnal cycle) and mean sea level pressure for each experiment is given in Section 3. A detailed discussion and physical interpretation of the results shown in Section 3 are given in Section 4. Finally, the conclusions and future work / applications are given in Section 5.



## 2 Model and Methods

### 2.1 Model background

The general circulation model (GCM) used in this study is the atmosphere-only version of the Australian Community Climate and Earth System Simulator (ACCESS), which is described in more detail in Frauen et al. (2014) and Bi et al. (2013). ACCESS

is developed from the United Kingdom Met Office Unified Model (MetUM), Hadley Centre Global Environmental Model version 2 (HadGEM2: Martin et al., 2011) and has a horizontal grid spacing of 3.75° longitude by 2.5° latitude and 38 vertical levels in this study. Parameterized processes include clouds, precipitation, surface energy exchange, boundary layer processes and radiation.

Relevant to the experiments used in this study is the the surface processes parameterization, which is the Met Office Surface

Exchange Scheme (MOSES: Cox et al., 1999; Essery et al., 2001). Heterogeneity of the land surface is represented in MOSES by splitting the land into smaller tiles (i.e. sub-grid box scale). The tiles can be any combination (fractional) of nine different surface types, which are separated into five vegetated (broad leaf trees, needle leaf trees, two types of grasses and shrubs) and four non-vegetated (lakes, urban, bare soil and permanent ice) surfaces. The surface temperature, radiative, sensible and latent heat fluxes are calculated for each surface type individually and area-weighted grid-box values are calculated from those and

passed back into the model. There are also four vertical layers in the soil (at 0.1 m, 0.25 m, 0.65 m and 2.00 m depth) and snow cover is represented by a single layer. More details of the MOSES scheme used in ACCESS can be found in Kowalczyk et al. (2013). In all simulations listed in Section 2.3, both the soil moisture content and deep soil temperatures are prescribed from climatological values (and updated monthly) in order to minimise feedbacks that may arise from circulation and precipitation changes in these simulations. This soil moisture contraint is applied only for these "proof of concept" experiments (outlined

below) and can be removed (i.e. freely-varying soil moisture).

### 2.2 Calculating land surface temperatures

#### 2.2.1 Original calculation in ACCESS

This section gives an overview of the processes that are considered for calculating the surface temperature ($T_*$) in ACCESS in order to show where the model code has been changed (including the names of the subroutines). The calculations for surface

temperature are given in more detail by Essery et al. (2001); however, this section only describes the equations that are changed (see Section 2.2.3) to prescribe $T_*$. Firstly, the surface energy budget equation is calculated as:

$$C_c \frac{dT_*}{dt} = R_N - H - \lambda E - G_0 \tag{1}$$

Where $C_c$ is the areal heat capacity of the surface (J m$^{-2}$ K$^{-1}$), $dT_*$ is the surface temperature increment (K), dt is the time step increment (s), $R_N$ is the net downward surface radiation (both long-wave and short wave, W m$^{-2}$), H is the downward

sensible heat flux (W m$^{-2}$), $\lambda E$ is the downward latent heat flux (W m$^{-2}$) and $G_0$ is the downward ground heat flux (W m$^{-2}$). A schematic of the model process for updating the surface temperature and fluxes is shown in Figure 1. Initially the surface



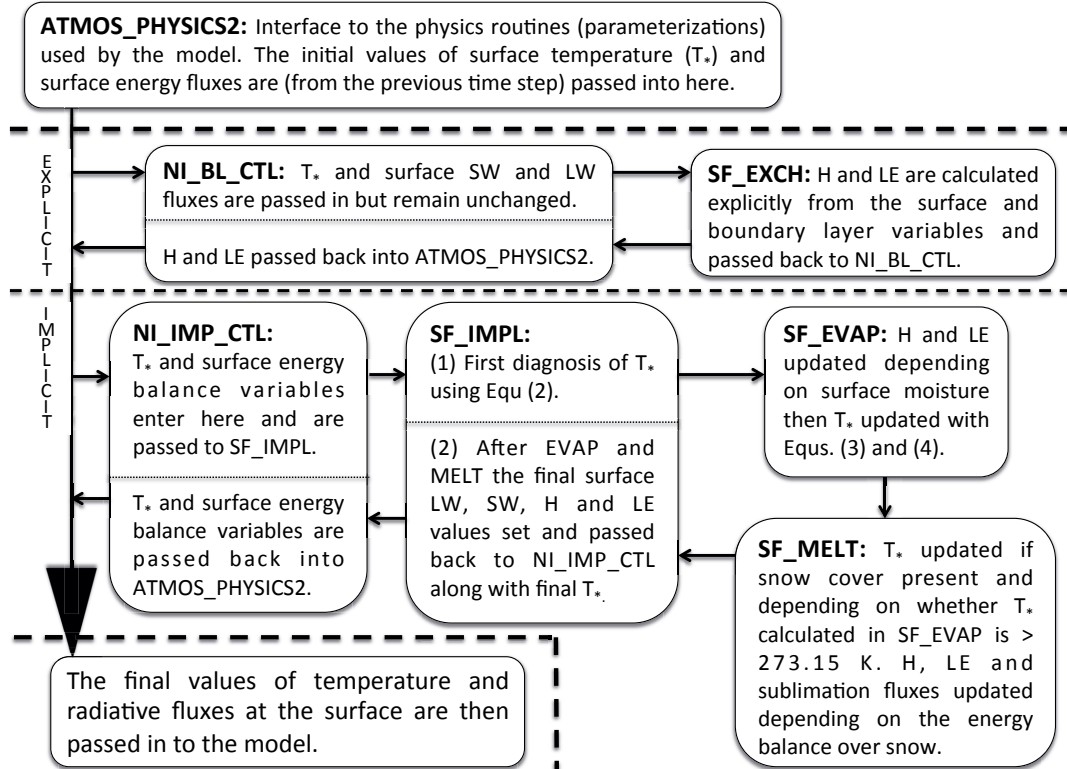

**Figure 1.** Schematic diagram of the processes involved with calculating the surface temperature and fluxes in ACCESS. UPPER CASE lettering refers to the names of individual subroutines within the model. The variables are passed from ATMOS_PHYSICS2 through the explicit calculations, then the implicit calculations and finally back to ATMOS_PHYSICS2 for use elsewhere. Arrows indicate the transfer of variables through subroutines. Dashed lines separate the transfer of variables into and out of the same subroutine where applicable.

fluxes are calculated explicitly at the start of a time step (in SF_EXPL, see Fig. 1) with temperatures from the previous time step (surface, soil and boundary layer). The fluxes are then updated implicitly, at which point, the initial estimate of the new value of surface temperature is calculated from:

$$T_* = T_s + \frac{1}{A_*}\left[ R_s - H - \lambda E + \frac{C_c}{\Delta t}\left( T_*^{(n)} - T_s \right) \right] \qquad (2)$$

5    Where $T_s$ is the temperature of the first soil layer beneath the surface (K), $A_*$ is the coefficient to calculate the surface heat flux (W m$^{-2}$ K$^{-1}$), $\Delta$t is the time step length (s), $T_*^{(n)}$ is the surface temperature from the previous time step (K), all other variables have the same definition as in Equ. (1). Adjustments to the surface sensible and latent heat fluxes are then calculated implicitly in SF_EVAP depending on the availability of surface moisture (Essery et al., 2001). The value of $T_*$ calculated in Equ. (2) then needs to be adjusted by an amount that is consistent with (and proportional to) the updated values of the sensible





and latent heat fluxes via:

$$\Delta T_{*EVAP} = -\frac{\Delta H + \Delta(\lambda E)}{A_*} \qquad (3)$$

$$T_{*EVAP} = T_* + \Delta T_{*EVAP} \qquad (4)$$

Where $\Delta T_{*EVAP}$ (K) is the land surface temperature increment resulting from the adjustments to the sensible ($\Delta H$) and latent heat ($\Delta\lambda E$) fluxes (W m$^{-2}$) and $T_{*EVAP}$ (K) is the adjusted value of land surface temperature following evaporation ($T_*$ and $A_*$ have the same definition as those in Equ. (2)). If there is no snow present within the grid box then $T_{*EVAP}$ is that final value of land surface temperature ($T_{*final}$, K) and is passed back in to the ATMOS_PHYSICS2. If there is lying snow however, then $T_{*EVAP}$ is passed into the SF_MELT routine (Fig. 1) to account for any melting ice and snow on land tiles.

The surface energy fluxes over snow and ice (sublimation and sensible heating) are also adjusted in SF_MELT. If the value of $T_{*EVAP}$ from (4) is above freezing for water ($T_m$, 273.15 K) then the temperature is adjusted by a value $\Delta T_{*MLT}$ (K), which is either:

1. back to freezing if there is sufficient snow that it cannot be melted within a time step (30 minutes in this case) or

2. by an amount proportional to the energy required to remove all the snow on the tile if it can all be removed within a time
step.

The final value of surface temperature that the atmosphere uses in the rest of the time step ($T_{*final}$, K) is therefore given as:

$$T_{*final} = T_{*EVAP} + \Delta T_{*MLT} \qquad (5)$$

If there is no melting then $\Delta T_{*MLT}$ is zero but if melting does occur then the surface fluxes are updated by an amount proportional to the value of $\Delta T_{*MLT}$. Therefore, the value of $T_*$ may differ *within* the model time step between the first guess
(Equ. (2)) and the final value (Equ. (5)), which also applies to the surface fluxes (H, LE and sublimation flux).

### 2.2.2 Creating the input surface temperature field

Given that ACCESS uses a thirty minute time step, in order to prescribe the land surface temperatures, a dataset that is available for all surface tiles and at thirty minute intervals is required. Such a dataset does not exist in the observational record and so therefore, in order to represent both the diurnal and seasonal cycles, the optimal solution is to take the surface temperatures
from a simulation where they are allowed to vary freely. In this study, surface temperatures are taken from each time step and tile from a fifty year long simulation that uses prescribed climatological SSTs and sea ice fractions (denoted as FREE in Table 1). Data are stored from each time step and surface tile type so that the prescribed temperature field can account for:

1. the diurnal and seasonal cycles in surface temperature

2. the surface heterogeneity over land (i.e. temperatures on individual tiles).



Starting at 00:00:00 UTC on $1^{st}$ January, all 50 values for that specific time produced by the FREE simulation (i.e. one for each year) are averaged together to produce a representative mean temperature on each land tile and saved. The process is then repeated on all land tiles for 00:00:30 UTC on $1^{st}$ January. The process is repeated for all time steps over the year to produce a climatological land temperature field that contains a mean diurnal cycle for each day of the year on each land surface tile.

This is illustrated in Fig. 2 for a selection of different grid points in the model. These grid points are located within a tropical (Amazonia), sub-tropical (central Australia), high-latitude (northern Asia) and mid-latitude (Europe) region. The grey lines show the thirty minute surface temperatures at those points for all fifty years of FREE on the $1^{st}$–$2^{nd}$ January and the black solid line is the average over those fifty years for each thirty minute time step (Fig. 2, middle column). The variability in surface temperatures is reduced by taking the average; however, a clear diurnal cycle can be seen at each of those grid points, which is

larger in the tropics than at mid-latitudes.

In Fig. 2, third column, all surface temperatures for all time steps in all years of FREE are plotted (grey) along with the mean diurnal cycle for each day (black) and the daily mean surface temperature (yellow). Again, it is clear that the mean diurnal temperature range is smaller than the full range of variability in surface temperature for each time step of all years, but there is a clear seasonal and diurnal cycle that is representative of the FREE simulation at each of those selected grid points.

Initial test experiments with the time step data resulted in two problems:

1. The time step (30-minute) dataset is too large to be read into the current ACCESS framework as one single input field.

2. Surface air temperatures (1.5 m above the surface) over the Antarctic reduced by >2 K relative to FREE.

To combat the first problem, surface temperatures are read into the model every three hours (when the radiative fluxes are updated) and interpolated hourly between those points (orange line overlaid in Fig. 2, middle column). The results of the 30-

minute and 3-hourly temperature simulations have almost indistinguishable mean climate states (not shown). Therefore, the 3-hourly data are used in the simulations outlined below.

In order to prevent the atmosphere temperatures above the Antarctic from reducing significantly, the surface temperatures on permanent land ice tiles were allowed to vary freely. The impact of this exception is small and discussed in Section 4.

### 2.2.3 Implementing the climatological land temperature dataset

In order to prescribe the land surface and sea ice temperature, Equ. (2) in SF_IMPL (Fig. 1) is simply changed to:

$$T_* = T_{pres} \tag{6}$$

Where T$_{PRES}$ is the input, prescribed temperature (K) field. Furthermore, the increments to the surface H, LE, sublimation and snow melt are still calculated in SF_EVAP and SF_MELT (Fig. 2) but the surface temperature increments (Equs. (4) and (5)) are removed so that the surface temperature cannot change. The final values of the surface radiation budget (right hand

side of Equ. (1)) are then set to their final values, which depend upon T$_{PRES}$ only.



**Table 1.** A list of the experiments run with ACCESS. The SST and sea ice fractional cover are climatological mean values representative of 1961-1990.

| Simulation (abbreviation) | Run length (years) | Land surface temperatures | Ice cover and SST | Perturbation to land temperature |
|---|---|---|---|---|
| Free-running (FREE) | 50 | freely evolving | prescribed 12-month periodic climatology | none |
| Control run 1 (CON1) | 50 | prescribed 3-hr interpolating climatology | as in FREE | none |
| Control run 2 (CON2) | 50 | as in CON1 | as in FREE | none |
| Heat all land (ALL10K) | 50 | as in CON1 | as in FREE | +10 K over all land points |
| Heat Amazonia (AMA10K) | 50 | as in CON1 | as in FREE | +10 K over all Amazonian land points |
| Heat the Maritime Continents (MC10K) | 50 | as in CON1 | as in FREE | +10 K over all Maritime Continent land points |
| Heat Australia (AUS10K) | 50 | as in CON1 | as in FREE | +10 K over all Australian land points |
| Heat North America (AM10K) | 50 | as in CON1 | as in FREE | +10 K over all North American land points |
| Cool North America (AMm10K) | 50 | as in CON1 | as in FREE | -10 K over all North American land points |

## 2.3 Experiments

The full list of experiments considered in this study are outlined in Table 1 along with the abbreviations used in the rest of this paper. A more detailed description of each experiment is given below. The following experiments are designed to either create the data necessary to prescribe the land surface temperatures or use those data. These first four experiments represent a suite of control simulations.

1. FREE: This simulation uses prescribed, climatological SSTs and sea ice fractions (monthly mean, 1961-1990 values) but allows the land temperatures to vary freely. The surface temperature from each surface type are used in each of the subsequent experiments below. This is denoted at the 'free running' (FREE) simulation.

2. CON1: Control run number 1, which same as FREE except the surface land temperatures are prescribed using the dataset described in Section 2.2.2.

3. CON2: Control run number 2, which is identical to CON1 except different initial conditions are used for the atmosphere.



Perturbation experiments are described in the following list where the surface state is changed by either heating (+10 K) or cooling (-10 K) the surface land temperatures over specific areas. The value of 10 K is intentionally chosen in order to induce a large and visible response in the atmosphere and not because such perturbations are based on actual observations (i.e. these are purely sensitivity experiments). If the resulting circulation responses are consistent with known physical processes then this is

indicative that the surface temperatures are being specified in the correct way. These perturbation experiments are:

4. ALL10K: Identical to CON1 except all land surface temperatures are increased by 10 K. This simulation is used to illustrate how the global circulation responds to an artificial enhancement of the land-sea thermal contrast.

5. AMA10K: The same as CON1 except the land temperatures within the box 285°E–310°E and 5°N–17.5°S are increased by 10K. This simulation is run to identify the seasonal and hemispheric impacts of heating Amazonia.

6. MC10K: The same as CON1 except the land temperatures within the box 100°E–160°E and 10°N–10°S are increased by 10K. This simulation is run to identify the seasonal and hemispheric impacts of heating the land within the West Pacific warm pool.

7. AUS10K: Identical to CON1 except surface temperatures are increased by 10 K over Australia. This is to identify the impact of land surface heating on the Australian monsoon and the Southern Hemisphere (SH) extratropical circulation.

8. AM10K: Identical to CON1 except surface temperatures over the North American continent are increased by 10K. This simulation is run to identify the impact of heating a large Northern Hemisphere (NH) continent on the extratropical circulation.

9. AMm10K: Identical to CON1 except surface temperatures over the North American continent are reduced by 10K. This simulation is run to identify the impact of cooling a large NH continent on the extratropical circulation.

## 3   Results

### 3.1   Surface air temperature at 1.5 m ($T_{1.5}$)

The differences in $T_{1.5}$ between CON1 and FREE are plotted in Fig. 3(a). The CON1 simulation is cooler over the Arctic by 0.1–0.5 K between 60°E and 60°W and warmer by 0.1–0.25 K over parts of Africa. Elsewhere, $T_{1.5}$ differences between CON1 and FREE are typically within ±0.1 K (i.e. small) and not statistically significant. There are also slight differences

between $T_{1.5}$ values in CON2 relative to CON1 (for example over both poles, Fig. 3(b)); however, those differences are not statistically significant and indicates that CON2 and CON1 are climatologically indistinguishable.

Increasing the prescribed surface temperatures on all land points (ALL10K) acts to significantly increase $T_{1.5}$ by more than 2.0 K (and more than 8.0 K over northern Asia) over all land surfaces (Fig. 3(c)) relative to CON1. There are also increases in $T_{1.5}$ of >2.0 K over the Arctic adjacent to the continents. Furthermore, $T_{1.5}$ values are significantly higher over the eastern

Pacific, north-west Atlantic, eastern Indian Ocean and parts of the Southern Ocean. Nevertheless, the largest changes in $T_{1.5}$





are primarily over the land surface with only small temperature changes (typically within $\pm0.5$ K) over the ocean where SSTs are unchanged (i.e. the same as in CON1).

In both of the tropical heating cases (AMA10K and MC10K), the largest increases of $T_{1.5}$ (relative to CON1) are restricted to Amazonia and the islands of the Maritime Continent (Figs. 3(d) and (e), respectively); however, there is evidence of the

atmosphere responding remotely from the low-level heating. For example, there are alternating positive and negative $T_{1.5}$ anomalies to both the north-east and south-west of the Amazon (Fig. 3(d)). In MC10K, similar (but weaker) alternating positive and negative $T_{1.5}$ anomalies extend to the north-east and south-east of the Maritime Continent too (Fig 3(e)).

In the AUS10K simulation, $T_{1.5}$ is higher over the Australian continent relative to CON1 (Fig. 3(f)). Despite the strong local heating the only significant remote responses are weak warming (0.1–0.25 K) over the Southern Ocean between $0°$–$60°$W and

weak cooling (0.1–0.5 k) over Antarctica.

The increases in $T_{1.5}$ for AM10K are largest over North America (Fig. 3(g)) and there is also evidence of increased temperatures (0.1–1.0 K) to the west of the continent (similar to ALL10K—compare Figs. 3(c) and (g)). There are also higher values of $T_{1.5}$ over the Arctic, central Asia and the Sahara that are statistically significant, which again indicates that there is a remote response to heating North America. In the AMm10K experiment almost the opposite is true. $T_{1.5}$ values are lower over North

America, the Arctic and the western Atlantic Ocean (Fig. 3(h)). Moreover, there is cooling over central Asia (approximately 0.1–0.5 K), albeit weaker than the warming induced in AM10K (compare Figs. 3(g) and (h)).

### 3.2 Precipitation

#### 3.2.1 Regional annual mean precipitation

The differences in the annual mean precipitation between CON1 and FREE are generally within $\pm8\%$ (Fig. 4(a)). The largest

percentage differences primarily occur over the Arctic circle (reductions $>4\%$) and the Amazon (increases $>4\%$). Nevertheless, the differences in precipitation outside these two regions (Arctic and Amazon) are largely statistically insignificant. Furthermore, for CON2 relative to CON1 (Fig. 4(b)) there are only small and non-significant differences in precipitation (within $\pm8\%$), which suggests that there is little impact on precipitation from changing the initial conditions.

For ALL10K relative to CON1 there are statistically significant changes to the precipitation over all land areas (Fig. 4(c));

however (unlike with $T_{1.5}$) the differences are not all the same sign. Precipitation increases by more than 30% over northern South America, Africa, south-east Asia, the islands of the Maritime Continent and, northern and eastern Australia but is reduced by more than 30% over central North America, central Asia and India. There are also large reductions (greater than 30%) in precipitation over the central Atlantic Ocean, Indian Ocean and much of the Pacific Ocean while there is an approximate 10% increase in precipitation throughout the Southern Ocean.

In both of the tropical heating cases (AMA10K and MC10K), precipitation increases by $>50\%$ where the surface temperatures are increased (compare Figs. 4(d) and (e) with Figs. 3(d) and (e), respectively). There are also precipitation anomalies of alternating sign that extend from the Amazon and the Maritime Continent to the north-east and south-west that are statistically significant (similar to the $T_{1.5}$ differences—Figs. 3(d) and (e)), which suggests the tropical heating is affecting precipitation re-

(c) Author(s) 2016. CC-BY 3.0 License.



motely (Fig 4(d) and (e)). Moreover, the response of tropical precipitation in AMA10K over Africa, India, the tropical Atlantic and Pacific is much stronger than in MC10K (the largest differences are confined to the west Pacific in MC10K).

Heating Australia causes precipitation to increase in the north and east of the continent but to decrease over the eastern Indian Ocean (Fig 4(f)). There is very little significant change in the precipitation field away from the Australian continent and eastern Indian Ocean.

For AM10K, increases precipitation coincide with the surface heating except in the centre of the continent (this also occurs in ALL10K—compare Figs. 4(g) and (c)). There is also higher precipitation over the Arctic and Greenland. Conversely, there is lower precipitation in the Gulf of Mexico and the East Pacific. For AMm10K, there is a reduction in precipitation throughout North America, which extends over Greenland and into the Arctic (Fig. 4(h)). There are also significant increases in precipitation over the North Atlantic and the North Pacific with reduced precipitation over North Africa.

### 3.2.2 Diurnal cycle in the tropics

When prescribing the surface temperatures it is important to maintain the diurnal cycle, particularly in regards to the impact of the daily heating and cooling of the land surface on tropical convection. Accepting that ACCESS and other GCMs produce convective rainfall too early in the day relative to observations (Yang and Slingo, 2001; Dai and Trenberth, 2004; Dai, 2006; Dirnmeyer et al., 2012), including ACCESS (Ackerley et al., 2014, 2015) this should also occur in the prescribed simulations outlined in Section 2.3. Nevertheless, the model needs to representative of the free-running simulation and therefore the early triggering of convective rainfall is expected. In order to asses this, the mean diurnal cycle of convective rainfall is plotted in Fig. 5 for tropical land grid points in:

1. West Africa, 0°E, 15°N (June-July-August mean for a NH monsoon region), Fig. 5(a).

2. North Australia, 135°E, 15°S (December-January-February mean for a SH monsoon region), Fig. 5(b).

3. The Maritime Continent (Borneo), 112.5°E, 0° (annual mean for an equatorial island), Fig. 5(c).

4. Northern South America (central Amazonia), 300°E, 0° (annual mean for an equatorial mid-continent point), Fig. 5(d).

In West Africa (Fig. 5(a)), convective rainfall peaks around 1030 Local Time (LT) in FREE. Both CON1 and CON2 have peak rainfall around 1030–1330 LT with higher rainfall between 1330–1930 LT. Despite these differences the diurnal cycle of rainfall is *maintained* in both CON1 and CON2.

Convective rainfall in northern Australia peaks at 1100 LT in FREE, CON1 and CON2; however, as over West Africa, the prescribed simulations have higher precipitation in the afternoon (around 1700 LT). Despite the higher rainfall around 1700 LT, the diurnal cycle still occurs in the prescribed simulations. Interestingly, the secondary peak in rainfall (around 0200 LT) associated with the modelled diurnal cycle of the heat low circulation (as discussed by Ackerley et al., 2014, 2015) is represented in each of the prescribed simulations. This suggests that the diurnal cycle of the low-level atmospheric circulation at this point is also maintained in CON1 and CON2.



For the Maritime Continent (Fig. 5(c)), the peak in convective rain occurs at 1130 LT in all simulations; however, the rainfall amounts are slightly higher in CON1 and CON2. Moreover, the afternoon rainfall is slightly higher in CON1 and CON2 relative to FREE (as with northern Australia and West Africa), but the overall diurnal cycle is maintained (including the secondary peak around 0230 LT).

Finally, peak convective rainfall occurs at 1330 LT in all simulations for the Amazonian point (Fig. 5(d)); however, CON1 and CON2 both have higher accumulated precipitation than the FREE simulation between 0730–1930 LT, which agrees with the region of increased annual mean precipitation in Fig. 4(a). Nevertheless, the diurnal cycle in convective precipitation is again maintained in both CON1 and CON2 when the temperatures are prescribed as it is in the other tropical regions.

### 3.3   Mean sea level pressure

The differences in mean sea level pressure (MSLP) between FREE and CON1 (Fig. 6(a)) generally lie within ±0.5 hPa of each other across the globe and are not statistically significant. Similarly, for CON2 relative to CON1 (Fig. 6(b)) the differences in MSLP are not statistically significant across almost all of the globe.

The largest differences in MSLP occur in the ALL10K experiment with reductions of 0.5 to 2.0 hPa over most global land surfaces, the Atlantic Ocean, the Arctic and the Southern Ocean between 180°W and 30°W (Fig 6(c)). There are increases
in MSLP of 0.5 to 8 hPa over the North Atlantic, North and South Pacific, and the Southern Ocean between 20°W to 180°E. Heating the global land surface is therefore having a large impact on the whole global circulation and is not just restricted to over the land.

There are also significant changes in global MSLP in both the AMA10K and MC10K simulations. The MSLP reduces over the Amazon by more than 4 hPa in AMA10K with reductions of more than 0.5 hPa over much of the Atlantic Ocean (Fig.
6(d)). Over the Maritime Continent MSLP is only lower by approximately 0.5 hPa (Fig. 6(e)). Despite the weaker local MSLP response in MC10K relative to AMA10K, both simulations have statistically significant MSLP anomalies (of alternating sign) that extend from the tropics into the mid-latitudes, which suggests that there is also a remote circulation response to the tropical heat sources.

In the AUS10K experiment (Fig 6(f)), there is a reduction in MSLP over the Australian continent from the surface heating;
however, there are also statistically significant increases in MSLP over the Southern Ocean and decreases over the Antarctic. Heating the Australian continent therefore appears to affect both the SH mid-to-high-latitude and the local continental scale circulations.

Similarly, heating and cooling North America has a large impact on the NH mid-latitude circulation. Heating North America reduces the MSLP locally by 0.5–2.0 hPa but there is also lower MSLP over western Europe (Fig. 6(g)). Conversely, the MSLP
is 0.5-2.0 hPa higher over eastern Asia and the North Pacific. When the North American continent is cooled (AMm10K) the MSLP increases locally by 0.5-2.0 hPa (also over Greenland) with lower MSLP (again 0.5-2.0 hPa) over east Asia and the North Pacific (Fig. 6(h)).



## 4 Discussion

### 4.1 Control experiments

#### 4.1.1 FREE vs CON1

Over most of the globe, the differences in $T_{1.5}$ between FREE and CON1 are within $\pm0.1$ K (unshaded in Fig. 3(a)). Impor-

tantly, the differences of $T_{1.5}$ over the Antarctic in CON1 relative to FREE are not statistically significant (Fig 3(a)). Therefore despite allowing the surface temperatures to vary freely in CON1, the surface air temperatures over the Antarctic are unaffected as a result of prescribing the surface temperatures over all other land surface tiles. Nevertheless, there are some regions where $T_{1.5}$ is significantly different between FREE and CON1 for example over the NH high-latitudes (Fig. 3(a)). The largest difference in $T_{1.5}$ between CON1 and FREE (-1.32 K) occurs at 277.5°E (82.5°W) and 67.5°N (in northern Canada) and the

anomaly is particularly pronounced between September and May (and particularly in December to February—not shown). It is hypothesised that the prescribed surface temperatures in the CON1 simulation may be changing the surface snow cover relative to FREE over the NH high-latitudes.

To investigate this hypothesis, the snow mass at 277.5°E and 67.5°N during September, October and November (SON) is plotted in Fig. 7(a). The values for each individual day of SON are averaged over all 50 simulation years to give the mean time

series of snow accumulation in FREE (solid line) and CON1 (dashed line) during that season (Fig. 7(a)). From approximately day 29, the CON1 simulation has (on average) more snow lying on the surface than FREE (Fig. 7(a)), which continues into boreal winter (not shown). The prescribed surface temperatures in CON1 therefore, are causing more snow to accumulate relative to FREE and the reason for this can be seen in Fig. 7(b). The daily maximum surface temperature at 277.5°E and 67.5°N during SON in CON1 (black, solid line) is plotted in Fig. 7(b). The day on which the maximum surface temperature

drops below 0°C is denoted by the dashed lines and corresponds with day 29 (as also marked in Fig. 7(a)). After this point, the surface temperature does not rise above the freezing point of water and therefore the surface snow cannot melt away. Conversely, in many of the 50 realisations of SON in FREE (grey lines, Fig. 7(b)), the maximum surface temperatures remain above 0°C past day 29 of SON and so the snow can still melt after this point. Therefore, due to prescribing the surface temperatures, snow melt is typically prevented earlier in CON1 than FREE and so snow amounts are, on average, higher in

CON1 during the cold season, which causes $T_{1.5}$ to be systematically lower.

The reduced values of $T_{1.5}$ within the Arctic Circle appear to cause a reduction in precipitation westward of Greenland and the to the north-east of Asia; however, the differences in precipitation over the rest of the globe between CON1 and FREE are largely insignificant (Fig. 4(a)). Moreover, the differences in mean sea level pressure between CON1 and FREE are also largely insignificant (Fig. 6(a)). It appears that differences in $T_{1.5}$ between CON1 and FREE have relatively little impact on

the global precipitation and circulation fields. Therefore the prescribed land surface temperature simulation (CON1) is broadly able to reproduce the climate of the original simulation (FREE) from which the land surface temperatures are derived.



### 4.1.2 CON1 vs CON2

The differences in $T_{1.5}$ (Fig. 3(b)), precipitation (Fig. 4(b)) and mean sea level pressure (Fig. 6(b)) between CON1 and CON2 are climatologically indistinguishable. The climatological state of the modelled atmospheres in CON1 and CON2 are therefore not sensitive to changing the initial conditions and shows further that this model setup is reliable for performing the idealised
simulations in this study.

## 4.2 Temperature perturbation experiments

### 4.2.1 All land heating

Previous work by Chadwick et al. (2013b) shows that induced heating of the land surface causes an increase in tropical precipitation in GCM experiments with prescribed SSTs. Nevertheless, in order to induce that surface warming Chadwick et al.
(2013b) either quadrupled $CO_2$ concentrations or increased the solar constant, therefore the surface temperature response to those perturbations would have been unknown until after the experiments were run. The method of prescribing surface temperatures shown in this study therefore presents an opportunity to assess the impact of increasing land surface temperatures—by a pre-determined quantity—on tropical (and global) precipitation in comparison to those of Chadwick et al. (2013a) who increase land surface temperatures indirectly.

An increase in precipitation over almost all tropical land surfaces can be seen in the ALL10K experiment (Fig. 4(c)). To first order, the changes in precipitation appear to be caused by enhanced convection over the land (uplift) and suppressed convection over the ocean (subsidence), which coincide with a reduction in MSLP (Figs. 4(c) and 6(c)) as suggested by Bayr and Dommenget (2013). Nonetheless, the pattern correlation between the differences in precipitation and MSLP in Figs. 4(c) and 6(c) is weak (-0.20) and there are several regions where the MSLP and precipitation differences are the same sign (e.g.
over the Atlantic and central Asia). Therefore MSLP may not be a good indicator of the changes in circulation that are causing the changes in precipitation.

  The mean, pressure vertical velocity at 500 hPa ($\omega_{500}$) is plotted for CON1 in Fig. 8(a) with dashed lines indicating areas of climatological ascent and solid lines for subsidence. The same field is given for ALL10K in Fig. 8(b) (contours) with the difference in $\omega_{500}$ for ALL10K relative to CON1 overlaid (red indicating relative subsidence and blue relative ascent). There
is a strengthening and expansion of the ascent regions over central-southern Africa, northern South America, the islands of the Maritime Continent and northern Australia with increased subsidence over the tropical-sub-tropical Atlantic, Indian Ocean and the ocean surrounding the Maritime Continent. Moreover, the pattern correlation between the $\omega_{500}$ anomalies in Fig. 8(b) and the precipitation anomalies in Fig. 4 is -0.69, which indicates that $\omega_{500}$ is a better indicator of the circulation-induced precipitation changes than the MSLP. These results also agree with the results of Chadwick et al. (2013a), who show that the
spatial patterns of tropical precipitation response are also driven by circulation changes and not just the local thermodynamic influence (i.e. surface heating). While it should be expected that the largest changes in precipitation should be over the land (given the pattern of surface heating), precipitation does not increase over all land grid points. This is most apparent over



the Indian sub-continent where (to first order) the increased surface temperatures should enhance precipitation; however, the large-scale re-organisation of the tropical circulation (seen in Fig. 8) increases subsidence over India and suppresses rainfall.

### 4.2.2 Tropical Heating

In both the AMA10K and MC10K experiments, there is evidence of alternating $T_{1.5}$, precipitation and MSLP anomalies

emanating from the heating region and extending into the mid-latitudes of both hemispheres (see Section 3). These $T_{1.5}$, precipitation and MSLP anomalies that alternate in sign suggest that there are waves propagating away from the imposed tropical heat sources (Gill, 1980). Such a response is consistent with the modelling study of Hoskins and Karoly (1981) where low-latitude heating can excite Rossby wave propagation into the high-latitudes provided there was a background westerly flow. Moreover Hoskins and Ambrizzi (1993) and Jin and Hoskins (1995) showed that the excitement of Rossby waves from

a tropical source depends on the location of the heating and the background zonal flow in the tropics and mid-latitudes, which vary seasonally. In order to identify whether the $T_{1.5}$, precipitation and MSLP features are associated with wave propagation away from the tropics, the characteristics of the upper-level atmospheric flow need to be considered. Hoskins and Karoly (1981) and Hoskins and Ambrizzi (1993) primarily focus on the 300 hPa fields, which are also considered here for ease of comparison.

The differences in the zonal mean deviation of the 300 hPa streamfunction (contours) for AMA10K and MC10K relative

to CON1 are plotted in Fig. 9. The fields are time-averaged annually (ANN), for December–February (DJF) and for June-August (JJA). The orange boxes denote the land areas where the surface temperature has been increased by 10 K. In both the AMA10K and MC10K experiments (Figs. 9(a) and (d)), alternating positive and negative streamfunction anomalies can be seen emanating from the heating region and into the high latitudes of both hemispheres. The magnitudes of the streamfunction anomalies appear to be stronger in the AMA10K simulation than the MC10K simulation, which may be due to the smaller

areal extent of the Maritime Continent islands and therefore their impact on the atmospheric circulation. Nevertheless, Hoskins and Karoly (1981) and Hoskins and Ambrizzi (1993) show that if the heating anomaly is located in background easterly flow then this can suppress the development of waves that propagate towards higher latitudes. Regions where the 300 hPa mean flow is negative (easterly) are stippled in blue in Fig. 9. The surface temperature perturbations in the MC10K experiment lie completely within a region of background easterly flow whereas the AMA10K heating region extends into areas with background westerly flow in both hemispheres. Therefore the background atmospheric state is playing a role in weakening the

teleconnections between the tropical heating and mid-latitude circulation in the MC10K experiment relative to AMA10K.

The importance of the surface heating location relative to the background flow, rather than simply the areal extent of the heating source, is more obvious when the seasonal (DJF and JJA) averages are considered. In DJF (Figs. 9(b) and (e)), background easterly flow is located between 0°E–150°E and 5°N–10°S and over a small region of northern South America. As

the AMA10K heating zone extends into regions of westerly background flow in both hemispheres during DJF, there is strong wave activity in both the NH and SH (Fig. 9(b)) although the streamfunction anomalies are stronger in the winter hemisphere. As the Maritime Continent lies within climatological easterlies in the MC10K simulation, the waves appear weaker in the streamfunction field in both hemispheres although the waves are still present (Fig. 9(e)).



In JJA, the Amazonian heating source lies entirely south of the band of background easterly flow at 300 hPa and there is little wave activity apparent in the streamfunction field in the NH as a result (Fig. 9(c)). Moreover, there is a much broader band of background easterly flow northward of the Maritime Continent heating source and subsequently there is no evidence of wave activity propagating into the NH high-latitudes (Fig. 9(f)). There is however, strong wave activity in the SH during JJA in both the AMA10K and MC10K experiments (Figs. 9(c) and (f)), where the background westerly flow adjacent to the heating region allows for Rossby wave propagation into the higher latitudes. Therefore, based on the evidence given above, it is more likely to be the background atmospheric state, as opposed to the areal extent of the surface heating, that is causing the stationary Rossby waves in each hemisphere.

Overall, the circulation response to both of these tropical heating sources are consistent with the results of Hoskins and Karoly (1981), Hoskins and Ambrizzi (1993) and Jin and Hoskins (1995). Therefore, the idealised GCM with prescribed land surface temperatures in this study is likely to be useful for running similar experiments.

### 4.2.3 Sub-tropical heating: Australia

Previous work has shown that Australian rainfall has changed regionally over the last 60 years (Smith, 2004; Berry et al., 2011); however, there has only been one study that perturbed the local surface conditions over the continent in order to account for the changes (Wardle and Smith, 2004). Wardle and Smith (2004) reduced the land surface albedo by a factor of four over the whole of the Australian continent to induce a surface warming and cause an increase in monsoon rainfall. The AUS10K experiment (Table 1) now provides an opportunity to compare the impact of directly heating the Australian land surface with an indirect heating (i.e. reducing the surface albedo as in Wardle and Smith, 2004).

Precipitation increases are primarily in the north and east (Fig. 4(f)), which implies that the monsoon driven rainfall is responding the strongest as the largest changes occur in DJF (not shown). Moreover, the increase in precipitation is primarily through increased convective precipitation, which suggests an increase in ascending air over the continent, which causes the MSLP to be lower over Australia (Fig. 6(f)). Reduced MSLP and increased monsoon rainfall also occur with reduced surface albedo (Wardle and Smith, 2004) and shows the increased surface temperature in AUS10K is having a similar impact.

The change in convective rainfall over Australia also appears to be driving changes in the SH mid-latitude circulation. MSLP increases by >0.5 hPa over the Southern Ocean and decreases by a similar magnitude over the Antarctic (Fig. 6(f)). The MSLP changes are consistent with a transition towards the positive phase of the Southern Annular Mode (SAM, Thompson and Wallace, 2000). Moreover, there is also a poleward shift in the annual mean location of the SH mid-latitude jet (Fig. 10(a)), which is consistent with a more positive phase of SAM. The largest changes in the zonal wind occur in DJF (0.5–2.0 m s$^{-1}$, Fig. 10(b)) rather than JJA (typically <0.5 m s$^{-1}$, Fig. 10(c)), which coincides with the periods where the Australian monsoon is active and inactive, respectively. Therefore the enhanced Australian monsoon appears to be strengthening the SH Hadley circulation as a result of increased convection over the continent.

Such an impact on the SAM was not discussed in Wardle and Smith (2004) and warrants further investigation—especially given that there has been a shift towards a more positive phase of the SAM in DJF over the last 60 years (Gillett et al., 2013). The majority of the trend towards a more positive SAM is attributed to SH stratospheric ozone depletion (Arblaster and Meehl,



2006; Polvani et al., 2011); however, greenhouse gases also play a weaker role in in the positive trend in the SAM index, which may in part be caused by an increase in the SH meridional temperature gradient (Arblaster et al., 2011). Given that land surfaces are expected to warm more than the ocean from increasing greenhouse gases (Sutton et al., 2007; Joshi et al., 2008; Dommenget, 2009), the model developed in this study could be used to understand the impact of the land-sea warming contrast

on large-scale modes of atmospheric variability (such as the SAM).

### 4.2.4  North American experiments

Heating (AM10K) and cooling (AMm10K) the North American continent induces local decreases and increases in MSLP, respectively (Figs. 6(g) and (h)). Moreover, precipitation increases over most of North America in AM10K (except the central plains, Fig. 4(g)) and decreases in AMm10K (Fig. 4(h)) in response to the heating and cooling, respectively. The largest changes

in precipitation occur in JJA (summer, not shown) where the increased surface temperature (Fig 11(a)) causes and increase in convective rainfall in AM10K (Fig 11(b)) and vice versa for AMm10K (Figs. 11(e) and (f)). It is also in JJA when the warm and cold anomalies in the annual mean $T_{1.5}$ over North Asia and North Africa (Figs. 3(g) and (h)) are at their strongest (Figs. 11(b) and (f)). Therefore, the rest of this section will focus on the changes in the JJA circulation in response to the surface heating and cooling anomalies imposed on the North American continent.

Locally, the surface heating and induced convection acts to reduce the surface MSLP in AM10K (relative to CON1), which can also be seen as a negative 850 hPa geopotential height ($Zg_{850}$) anomaly over North America (Fig 11(c)) and an associated anomalous cyclonic flow over the continent. Conversely, the $Zg_{850}$ field is higher in AMm10K than CON1 over North America and is associated with anomalous anticyclonic flow (Fig. 11(g)) in response to the lower surface temperatures and suppressed convection. There are also large differences in the $Zg_{850}$ and 850 hPa wind field to the west of North America with

an anomalous anticyclone and positive $Zg_{850}$ values over the North Pacific in AM10K (Fig 11(c)) and negative $Zg_{850}$ values and cyclonic flow in AMm10K (Fig. 11(g)).

Brayshaw et al. (2009) use GCM simulations to show that the presence of land in the extratropics acts to increase the low level surface drag on the mid-latitude westerly flow. Moreover, when a band of higher topography is placed on the extratropical land mass perpendicular to the incident westerly flow, there is weakening of the mid-latitude westerly flow throughout the

troposphere in response to the topographically induced drag (Brayshaw et al., 2009). A similar effect is seen in the AM10K experiment, where the increase in convective precipitation (and therefore deep convection) acts as a form of frictional drag on the incident westerly flow, which can be seen in the upper level (300 hPa) zonal wind field (Fig 11(d)). The impact of convective drag is known to weaken the daytime wind field in thermally direct circulation systems such as heat lows (Racz and Smith, 1999; Spengler et al., 2005) and is also represented in GCMs—including ACCESS (Ackerley et al., 2014, 2015;

Allcock and Ackerley, 2016). Therefore, the strengthening of convection in AM10K during JJA acts to decrease the strength or the upper level wind through increasing the frictional drag on the atmosphere in an analogous way to the dry convection in a heat low (Racz and Smith, 1999; Spengler et al., 2005) or by including orography (Brayshaw et al., 2009). Conversely, the reduction of convection acts to reduce the drag on the westerly flow in the AMm10K simulation and the zonal flow strengthens



at 300 hPa (Fig. 11(h)). It is the modulation of the mid-latitude wind field through enhancing or diminishing the "convective drag" that results in the changes in the circulation remotely from North America.

Brayshaw et al. (2009) also found that weakening of the upstream flow associated with increasing the surface drag is associated with a westward propagating Rossby wave. Such wave propagation is consistent with the $Zg_{850}$ anomalies in both
the AM10K and AMm10K simulations (Figs. 11(d) and (h), respectively. The upstream positive $Zg_{850}$ anomaly extends in to East Asia and induces anomalous east-to-southeasterly flow, which causes the increases in $T_{1.5}$ over North Asia (Fig. 11 (b)). Conversely, the negative $Zg_{850}$ anomaly upstream of North America in AMm10K induces north-to-northwesterly flow over North Asia and the subsequent reductions in $T_{1.5}$ (Fig. 11(f)). Given that these changes in atmospheric circulation in the AM10K and AMm10K experiments are physically consistent with those of other studies (for example Brayshaw et al.,
2009) there is potential to develop more realistic temperature perturbation experiments to explore the impact of surface land temperature anomalies on the mid-latitude circulations in both hemispheres.

## 5   Conclusions and further applications

The aims of this paper are to present the method of prescribing land surface temperatures in a GCM and show that the resulting simulated climate state is comparable with a simulation that uses freely evolving land temperatures. Furthermore, the study
has clearly shown that the atmospheric response to surface heating and cooling perturbations agree with previous studies using idealised GCM simulations. The main conclusions from this study therefore are:

- It is possible to prescribe land surface temperatures in ACCESS and produce a simulated atmospheric state similar to that of a freely-varying land temperature simulation.

- The diurnal cycle in tropical convection is maintained in the prescribed simulations.

- Heating all land surfaces by 10 K generally increases (decreases) precipitation over the land (ocean); however, the local response is governed by the strengthening or weakening of existing circulations seen in the control run.

- Regional surface heating in the tropics induce the formation of stationary Rossby waves that are dependent on the location of the heat source and the background state atmospheric zonal flow.

- Heating the Australian continent causes an increase in monsoon rainfall and shifts the SH mid-latitude westerlies poleward.

- Heating and cooling the land surface of North America acts to either weaken (heating the land) or strengthen (cooling the land) the mid-latitude westerly flow.

The experiments in this study showcase some specific examples of the potential applications for simulations with prescribed land surface temperatures. Further experiments / applications that could be developed include:

1. Develop prescribed land surface temperature simulations that are compatible with the Community Atmosphere Biosphere
Land Exchange (CABLE Kowalczyk et al., 2013) and the Joint UK Land Environment Simulator (JULES Best et al., 2011) models. The CABLE and JULES models are used in the latest versions of ACCESS and the MetUM GCMs and the development of the simulations described in this study (i.e. using MOSES) should allow this method to be applicable to both of those modules.





2. Remove the soil temperature and soil moisture constraints. This will allow the soil moisture to respond freely to imposed surface temperature field.

3. The adjusted radiative forcing has previously been calculated in simulations with prescribed SST that allow the atmosphere and land surface to respond freely to changes in $CO_2$ (for examples see Andrews et al., 2012; Hansen et al., 2005). Nevertheless, Andrews et al. (2012) state that, "Land temperatures can, for example, respond in fixed SST experiments. This gives rise to a global temperature increase that may cause circulation changes and other responses that affect the radiation balance", which presents a limitation to their analysis. Shine et al. (2003) show that the radiative forcings caused by $CO_2$, aerosol and ozone perturbations in simulations with both prescribed land and sea surface temperatures were an "excellent indicator of the surface temperature response" in parallel simulations using a mixed-layer ocean and freely-varying land surface temperatures. Therefore, the ACCESS simulation with prescribed surface temperatures could be used for calculating the radiative forcing of $CO_2$ and aerosol in the same way as Shine et al. (2003) and reduce the circulation feedbacks noted in Andrews et al. (2012).

4. AMIP simulations with perturbed SSTs (e.g. uniform increase of global SST by 4K—AMIP4K) and greenhouse gases (quadrupled $CO_2$ with prescribed AMIP SST—AMIP4x$CO_2$) are available in the CMIP5 archive; however, the simulations developed in this paper could be used to develop an AMIP simulation with all surface temperatures increased uniformly by 4 K (e.g. AMIP4K$_{all}$) with and without $CO_2$ perturbations. Furthermore, there is also the potential for running coupled atmosphere-dynamical ocean simulations with prescribed land surface temperatures (reverse-AMIP i.e. freely evolving ocean, prescribed land). Such simulations would reveal the impact of coupled ocean-atmosphere circulation errors that result from biases in the representation of land surface temperatures.

5. Three-hourly surface temperature data are available from other CMIP5 models (apart from just ACCESS). Therefore, given the method described in this paper, those other models' surface temperature fields could be applied to ACCESS in order to identify whether the circulation biases in individual CMIP5 models are driven by errors in their surface temperatures (i.e. if circulation errors are surface temperature driven then they should occur when applied to ACCESS).

6. Instead of holding the surface temperature to a fixed value the approach can be altered by adding a flux correction term to the surface temperature tendency equation (Sausen et al., 1988). This is a common approach in coupled GCM model development to correct SSTs in simplified or biased ocean models (for example see Collins et al., 2006). Such a method would allow the flux correction to be applied to the full global surface (and not just the ocean-atmosphere interface).

While this list is not exhaustive, it presents some logical steps forward for further testing and development.



## 6 Code availability

The model source code for ACCESS is not publicly available; however, more information can be found through the ACCESS-wiki at: $https://accessdev.nci.org.au/trac/wiki/access$. Any registered ACCESS users who wish to gain access to the source code described in this paper can do so from:

5    $https://access-svn.nci.org.au/svn/um/branches/dev/dxa565/src\_presT\_reg/src@9826.$

*Acknowledgements.* This project was funded by the ARC Centre of Excellence for Climate System Science (CE110001028). The ACCESS simulations were undertaken with the assistance of the resources from the National Computational Infrastructure (NCI), which is supported by the Australian Government.



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





**Figure 2.** Examples of how the surface temperature (K) inputs were produced at individual grid points. Left column: the locations of the example grid points. Middle column: Corresponding surface temperature values for those points in the left column on $1^{st}$ and $2^{nd}$ January. Grey lines are the surface temperatures for each of the 50 years and the black lines represent the time-step mean (30 minute) values from those 50 years on $1^{st}$ and $2^{nd}$ January. Right Column: All surface temperature values at the grid points specified in the left column from the 50-year FREE simulation (grey lines), the time-step mean values (black line) and the three-hourly input, hourly interpolated temperature field described in Section 2.2.2. The yellow line is the daily mean surface temperature (highlights the seasonal cycle).





**Figure 3.** Differences in annual mean surface air temperature at 1.5 m (K) for (a) CON1 - FREE, (b) CON1 - CON2, (c) ALL10K - CON1,
(d) AMA10K - CON1, (e) MC10K - CON1, (f) AUS10K - CON1, (g) AM10K - CON1 and (h) AMm10K - CON1. Values of p≤0.05 are
denoted with an x.





**Figure 4.** Differences in annual mean precipitation (%) for (a) CON1 - FREE, (b) CON1 - CON2, (c) ALL10K - CON1, (d) AMA10K - CON1, (e) MC10K - CON1, (f) AUS10K - CON1, (g) AM10K - CON1 and (h) AMm10K - CON1. Values of p≤0.05 are denoted with an x.







**Figure 5.** Diurnal cycle of convective precipitation in the tropics (mm 3hr$^{-1}$) at (a) 0°E and 15°N (West Africa) in JJA, (b) 135°E and 15°S (North Australia) in DJF, (c) 112.5°E and 0°N (Borneo, equatorial island) annual mean and (d) 300°E and 0°N (Amazon, equatorial continental) annual mean.





**Figure 6.** Differences in annual mean, mean sea level pressure (hPa) for (a) CON1 - FREE, (b) CON1 - CON2, (c) ALL10K - CON1, (d) AMA10K - CON1, (e) MC10K - CON1, (f) AUS10K - CON1, (g) AM10K - CON1 and (h) AMm10K - CON1. Values of p≤0.05 are denoted with an x.





**Figure 7.** Time series of (a) mean daily snow amounts in SON averaged over 50 years of simulation in FREE (solid line) and CON1 (dashed line). (b) Time series of maximum daily surface temperatures during SON from all years in FREE (grey lines) and CON1 (solid black line).



**Figure 8.** The climatological mean (averaged over all years of simulation) pressure vertical velocity at 500 hPa ($\omega_{500}$, Pa s$^{-1}$) in the (a) CON1 and (b) ALL10K simulations. Solid lines indicate positive (subsidence) and dashed lines negative (uplift) values. Overlaid in (b) are the differences between ALL10K and CON1 where red shading indicates a positive difference and blue shading negative.





**Figure 9.** Differences in the deviation of the zonal mean streamfunction at 300 hPa between AMA10K and CON1 for (a) annual mean, (b) DJF mean and (c) JJA mean, and between MC10K and CON1 for (d) annual mean, (e) DJF mean and (f) JJA mean (contours). Orange boxes indicate the area where the land surface temperatures were increased by 10 K in AMA10K (top row) and MC10K (bottom row). Grid points where the mean background zonal flow is easterly are stippled in blue.





**Figure 10.** The difference in the 850 hPa zonal flow in AUS10K relative to CON1 for the (a) annual mean, (b) DJF-mean and (c) JJA mean (shaded). Overlaid (solid contours) is the mean zonal flow in CON1 to highlight the location of the westerly jet at 850 hPa.





**Figure 11.** The JJA-mean differences between AM10K and CON1 simulations for (a) convective precipitation (%), (b) $T_{1.5}$ (K), (c) 850 hPa geopotential height and wind field and (d) the 300 hPa zonal wind (300 hPa zonal wind from CON1 oveerlaid with solid contours). The JJA-mean differences between AMm10K and CON1 simulations for (e) convective precipitation (%), (f) $T_{1.5}$ (K), (g) 850 hPa geopotential height and wind field and (h) the 300 hPa zonal wind (300 hPa zonal wind from CON1 overlaid with solid contours).