# Peer review of "Atmosphere-only GCM simulations with prescribed land surface temperatures."

_Geoscientific Model Development, 2016_

## Referee Comment (RC1) · R. Law (Referee) · 15 Feb 2016

General comments

This paper describes a method for prescribing land surface temperatures in an atmospheric model and then applies the method in a series of sensitivity tests. It is suitable for publication with minor revisions, although a restructure of the paper might make it an easier read (see specific comments).

Specific comments

Sec 2.2.2 and Figure 1: There appears to be a discontinuity at 0Z in the 1-2 January timeseries plots in the middle column of the figure. While other step changes appear commonly every 3 hours, presumably related to the radiation time-step, the 0Z step

appears more consistent/worse, at least for Australia and N Asia. For N Asia this becomes the dominant feature in the figure rather than any diurnal cycle (and so appears to contradict the statement that 'a clear diurnal cycle can be seen at each of those grid points' (p6, line 9)). While I don't expect this issue to have any implications for the work presented here, a comment/explanation in the text would be useful to satisfy a curious reader.

Sec 2.2.3: Are there any implications for the surface energy balance in prescribing the surface temperature, or is any implied imbalance absorbed in the radiation terms? Did you do any checks to confirm this?

Section 3: Please check references to east and west as they sometimes seem to be mixed up (see technical comments for examples).

Restructure of paper: There are two aspects to the paper. The first is checking whether the prescribed land surface temperature reproduces the original simulation and the second is the set of example sensitivity experiments. I think the paper would be easier to follow if the two aspects were dealt with separately in the results/discussion section, i.e. present all the 'CON1-FREE' and 'CON2-CON1' results first and discuss these before moving onto the presentation of the sensitivity experiments. These also might be best presented as groups of experiments with the results and discussion presented together for each group. It may just be personal preference, but I would find it easier to be able to look at the temperature, precipitation and pressure differences together for one experiment (or set of related experiments) before going on to consider the next experiment. If a restructure is undertaken, I would move the comment about Antarctic temperatures (p6, line 17 and line 22-23) into the results/discussion of 'CON1-FREE', e.g. 'Initial tests showed ....', 'This was resolved by ....' giving 'CON1-FREE' results as shown in Figure ...

Technical comments

p3, line 7: list Bi et al (2013) before Frauen et al. (2014) and perhaps note that from

the Bi et al. paper it is the ACCESS1.0 version that is most relevant.

p3, line 5: suggest adding 'configured similarly to ' before 'Hadley Centre ...'

p3, line 9: delete repeated 'the'

p3, line 19: 'constraint' mis-spelled

p5, line 8: delete 'the' before 'ATMOS_PHYSICS2'

p6, line 5: Might be worth noting that the grid-cell values shown are the mean across the tiles in the grid-cell, assuming that is the case.

p7, line 4: 'The first three experiments ...' not four.

p7, line 9: insert 'is the' before 'same'

p8, line 29 and 30: western Pacific, western Indian Ocean?

p9, line 6: south-east of the Amazon?

p9, line 10: the remote responses in the AUS10K temperature show some similarity to the con2-con1 differences. Do you think this is just coincidence?

p9, line 12: east of the continent?

p9, line 32: did you mean south-east, as this would be more consistent with the temperature anomaly?

p10, line 6: 'increased precipitation coincides'

p10, line 15: add 'be' before 'representative'

p10, line 17: 'assess' mis-spelled

p12, line 6: add 'Antarctic' after 'allowing the'

p12, line 26: delete space between T and 1.5

p12, line 27: delete 'the' at start of line

p16, line 10: 'an' not 'and' towards end of line

p16, line 13: Should be figures 11(a) and 11(e) not (b) and (f)

p16, line 31: replace 'or' with 'of'

p17, line 5: ')' after 'respectively'

p18, line 1: 'the' before 'imposed'

p20, line 3: 'GCM' instead of 'GCMs'

p20, line 21: 'model' not 'mode'

p21, line 4: 'of' repeated

p22, line 13: 'Cook' needs capital

Figure 1, middle column: the orange line is not defined in the figure caption (though the three-hourly input is mentioned for the right column but not shown?). Is the thickness of the black line significant or just for readability?

Figure 11: The colour bars in panels (a), (b), (e), (f) appear to be swapped.

―――――――――――――

---

## Referee Comment (RC2) · I. Watterson (Referee) · 16 Feb 2016

General Comments

In this paper, the authors argue that the impacts of land temperature anomalies on the atmosphere can be investigated by imposing constraints on an atmospheric GCM, in a similar way to simulations with sea surface temperature anomalies. A method of constraining land surface temperatures in the ACCESS model is presented and it is shown that the simulated climate, including the diurnal cycle, matches well the unconstrained result. A set of experiments, partly motivated by earlier studies, with land temperature in various regions changed by 10 K is then presented. These are of considerable interest and do provide a 'proof of concept'. The Discussion section then presents further results that explore physical mechanisms and make rather lengthy comparisons with
other studies. The final section makes some conclusions that seem overstated, and includes consideration of possible further experiments in unnecessary detail. In some respects, these sections go beyond the initial aim of the paper. Much of the presentation is very good and the work is potentially an excellent contribution. However, various limitations, noted below, also indicate a need for a considerable revision. Some reduction in the text could be needed, but some of the material might be better considered in a further paper in a different journal.

Specific comments (section or page-line given)

A. Prescribing land surface temperatures within a GCM could be a fairly simple exercise. In the case of ACCESS (2.2.1), the specification of surface temperature is evidently complicated, and the description given may not be well understood by a reader not familiar with the MOSES scheme. Eq 2 does not readily follow from Eq 1. It is not clear how 'surface' temperature relates to that of the first soil layer (of depth 0.1m), what G0 is and how it relates to T* and Ts. How does step 'n' relate to the final, etc?

B. Related to the specification of surface temperature anomalies should be a consideration of the energy fluxes associated with it. From P8L1 on, terms 'heating' and 'cooling' are used without explanation. Are these are the implied fluxes needed to keep a surface layer at the prescribed temperature? In any case, the surface (anomaly) must be then heating or cooling the air, which is clearly important. In fact, a warm surface might appear to be losing heat -so cooling, in that sense. Further description of these processes is needed.

C. The presentation of COM2 results (from P9L23 on) seems excessive. If the initial condition change is merely a tweak in the atmosphere, then one would expect no impact on the climate. Indeed there seems to be no statistically significant differences, so what is the interest in the CON2-CON1 results? (Presumably, they do indicate a typical pattern of random or weather-induced differences in 50y means.) A case could be made for averaging the two and using this as the base for other results. If not, some

amendment and reduction in the presentation can be made.

D. Regarding the tropically forced wave-like patterns (P14), while westerlies will aid propagation, other studies have shown that non-zonal components of a background state can also aid propagation through easterlies, especially into the winter hemisphere (which seems favoured in 9 b, c and f). Early studies include Schneider and Watterson (1984, J Atmos Sci) and Watterson and Schneider (1987, QJRMS), and these are built on more recently by studies such as Zhao et. al (2015, J. Climate). Could more recent studies than the three in 4.2.2 also be considered?

E. A potentially important result of the pair of AM experiments (P16) is that despite the large amplitudes (+10K, -10K) the response seems apparently linear, differing only in sign. Could this be highlighted? In any case, some of the discussion and comparison with earlier studies seems rather speculative. Does convection really act similarly to topography (P16L25)? Indeed, is there an explicit parameterization of the effect in the model? If not, what is the mechanism?

F. Despite rather extended discussions (section 4), the comparisons of the perturbed temperature cases with earlier studies can only be qualitative –the resulting temperature anomalies are different. The conclusions (P17L15) 'clearly show.. agree with previous studies' are rather over stated. Even at P1L12,'seems qualitatively consistent' might be enough. This links to the aims of the paper, as noted above.

Minor comments

P1L19 Land temperatures also respond to the simulated weather, of course.

P3L4 Since Bi describes two (coupled) versions, the one most like the model used here could be identified (presumably ACCESS1.0, as used in CMIP5, but at reduced resolution here).

P3L7 It might be more usual to state that 'Physical processes represented in the model include'. There are explicit components, in addition to some parameters.

[Figure]

P3L9 'the the'

P3L17 Does 'all' include FREE?

P3L17. Does 'deep soil' mean layers 2, 3, 4? Is there flux through the bottom of 4?

P4L1 Should this be 'SF_EXCH' –as in the Figure?

Fig 2 It seems the ' ..hourly interpolated temperature field' is in the middle column. The detail in the third column is not visible and seems to create an unwieldy file. It might be simplified.

P6L1,3 Does 00:00:30 mean 30 seconds after midnight? Should the first 00: be dropped?

P6 L17, 23 (and elsewhere) 'reduce' is being used in an uncommon, intransitive way
P6L27 'PRES', but (6) has lower case

Table 1. 'Maritime Continents'?

P7L8 'at the'?

P7L9 The soil temperatures and moisture are also prescribed, it seems.

P8L4 One might doubt if the processes in the response to such large (10K) anomalies can be known, from observations. Is this magnitude chosen to improve statistical significance of responses, given some expectation of linearity?

Fig. 3 Would grid square shading, as in Fig. 4, give a clearer depiction than the interpolated lines? Some explanation of the different usage could be added.

P8L22 (and later) If the 1.5m temperature is an interpolation from the surface (subject to parameterisations) then it will be strongly constrained to the prescribed land and sea values. Temperature at the first atmospheric level would be a stronger indication of an atmospheric response. A brief comment justifying the focus on 1.5m seems warranted.

P9L5,7 'alternating' does not seem a good description here –although it is better for

precip.

P9L10 lower case'k'

P10L16 'be representative'

P11L11 'Similarly' is odd. As before, CON2-1 is expected to be the same, but CON1-FREE is the main test.

P11L13 Presumably MSLP is an extrapolation from a surface that is now warmer, so one might expect it to be lower, even if the surface pressure is unchanged. How much of the lowering might be due to this? Is the surface pressure different?

P12L25 It seems the mean surface temperature is the same, but there tends to be more snow in CON1. How does that influence T1.5?

P13 Consistent with the earlier suggestion regarding CON1 and 2, this 4.1.2 seems unnecessary.

P13L9 Are the SSTs unchanged in Chadwick's warmer-land run?

P13L22. Would vertical velocity closer to the centre of the moisture column (e.g. 850 or 700hPa) be an even better match?

P14L2 'increases subsidence over India' is not clear.

P14L7 Often Rossby waves are excited by the latent heat from rain formation. Does this provide the 'imposed heat sources' that are described here? If so, does the reduced rainfall over the seas in MC10K counter the effect of enhanced rain over the land?

P16L10 'and increase'

Fig 11 labels are bulky –with m10K partly missing. The bars are incorrect (swapped) in a, b, e, f. Perhaps simplify, with bars combined for the pairs?

P17L21What supports 'the local response is governed by the strengthening .. of existing circulations'?

---

## Referee Comment (RC3) · N. Keenlyside (Referee) · 6 Mar 2016

Overall this is a well written paper introducing a very novel experimental design: prescribed land-surface temperature AGCM experiments. Some interesting but idealized experiments are also introduced to demonstrate that the approach gives reasonable responses. I recommend publication subject to some minor revisions, listed below.

Noel Keenlyside

Main concerns

(1) The description of the response to the NH heating, which seems not the most relevant. The study from Miyasaka and Nakamura (2005) is more relevant.

(2) I am not convinced about that there is a statistically significant response over the

SH westerlies induced by Australia heating.

Minor points

Pg2, L15, Without having read the entire paper, I find aim 2 a little hard to follow because you do not say that you prescribe the very same land surface temperature from the freely varying run, and that this implies that the experimental design does not introduce spurious effects.

Pg3, L15-20, I would have imagined that soil moisture would be a key variable to prescribe to the atmospheric model to capture the surface energy budget. I wonder what are the implications of fixing it to climatology in the 10K experiments? I think you should at least acknowledge that this might impact the results of the surface heating experiments. It might be worth mentioning here that snow cover is simulated? I wonder if you were to prescribe it, whether you would fix the deviations of CON from FREE

Pg3, L30, In my version latent heat is labelled here and in the equation as labdaE, while in the figure 1 it is LE.

Pg 8, s30, I am surprised that T1.5 does not heat further. It seems rather artificial that up to 8K temperature gradient can be formed in the lower 1.5 m of the BL. Some discussion is required of how this can be possible.

Pg 9, s20, Is there any reason to expect changes in the initial conditions should lead to a significant difference on these timescales?

Pg 10, s15, "including ACCESS" is misplaced.

Pg13, s5, Again, I am not clear why you would expect a difference between the CON1 and CON2 simulations. Memory of the atmospheric initial conditions is lost very quickly, and should be gone within a several months I think you should make this clear.

Pg 15, s5, while the arguments given seem reasonable, it seems hard to discount completely the extent of diabatic heating, which is surely greater the AMA case (as

seen in the precipitation field). I think you should be clear about this. Are the responses more comparable if they are scaled by the amount of diabatic heating?

Pg15, s30, The SH Hadley Cell should be present during JJA (i.e., strongest in the winter hemisphere). Are the changes in the SH winds statistically significant?

Pg 16, s5. It would be useful to put the anomaly surface heating into perspective. For example, could you please discuss it in terms of changes expected by the end of century? It would put the simulated responses into perspective.

S4.2.4, NA experiments. The mechanisms proposed by Brayshaw et al. (2009) are more relevant to the NH winter time circulation and the NA Storm track. I think the work of Miyasaka and Nakamura (2005) is much more relevant.

Takafumi Miyasaka and Hisashi Nakamura, 2005: Structure and Formation Mechanisms of the Northern Hemisphere Summertime Subtropical Highs. J. Climate, 18, 5046–5065.

In terms of the conclusions: (1) I think you should mention in the first bullet point "(excluding Antarctica)" , or something along those lines. Perhaps the reasons for this are not clear, and don't need to be explained as the experiments are still very interesting. (2) It is not clear to me that there really was a significant change in the SH circulation in response to Australian heating. (3) Also the explanation for the response to NA heating does not seem appropriate (see comment above).

———————————————

---

## Author Comment (AC1) · 21 Apr 2016

Reviewer general comments: This paper describes a method for prescribing land surface temperatures in an atmospheric model and then applies the method in a series of sensitivity tests. It is suitable for publication with minor revisions, although a restructure of the paper might make it an easier read (see specific comments).

Authors' response: The authors would like to thank the reviewer (Dr. R. Law) for her insightful, constructive and supportive review of our work. We have endeavoured to respond in detail to the comments raised and hope that we have answered those issues sufficiently. Please find them attached to this review as a supplement.

Specific comments Sec 2.2.2 and Figure 1: There appears to be a discontinuity at 0Z in the 1-2 January timeseries plots in the middle column of the figure. While other step

changes appear commonly every 3 hours, presumably related to the radiation time-step, the 0Z step appears more consistent/worse, at least for Australia and N Asia. For N Asia this becomes the dominant feature in the figure rather than any diurnal cycle (and so appears to contradict the statement that 'a clear diurnal cycle can be seen at each of those grid points' (p6, line 9)). While I don't expect this issue to have any implications for the work presented here, a comment/explanation in the text would be useful to satisfy a curious reader.

Response: The authors agree that this is a strange discontinuity and that it appears to be systematic in all 50 years of the simulation (grey lines in Fig. 2) for the Australian and North Asian points. We assume that it must be due to the radiation time step too. We also agree that the statement about a 'clear diurnal cycle' is misleading in this case and we have changed the text to be, "...however, diurnal variability in the surface temperature can be seen at..." in order to avoid saying 'diurnal cycle' specifically in reference to Fig. 2. We have also included the statement at the end of that paragraph to account for the discontinuity, which states that, "There are also some discontinuities in the original time step data, which are likely to be associated with the radiative calculations within ACCESS (occur every three hours)."

Sec 2.2.3: Are there any implications for the surface energy balance in prescribing the surface temperature, or is any implied imbalance absorbed in the radiation terms? Did you do any checks to confirm this?

Response: The responses of the surface fluxes are plotted in the attached FIG. 1 below. The differences in each of the fluxes are small (generally within $\pm 2.5$ W m-2) despite there being some statistical significance, which is not surprising given that over 50 years of simulation even small systematic differences are likely to be significant (although physically irrelevant in terms of the resultant climatological state e.g. global circulation). Given these small changes, the inclusion of these results in the main text will not change the conclusions or provide any more insight. Furthermore, as this figure will be published (and freely available) with the review responses, readers will be able

to view these figures (below).

Section 3: Please check references to east and west as they sometimes seem to be mixed up (see technical comments for examples).

Response: We thank the reviewer for noticing this and apologise for the systematic, unintentional misuse. We have corrected the text where necessary.

Restructure of paper: There are two aspects to the paper. The first is checking whether the prescribed land surface temperature reproduces the original simulation and the second is the set of example sensitivity experiments. I think the paper would be easier to follow if the two aspects were dealt with separately in the results/discussion section, i.e. present all the 'CON1-FREE' and 'CON2-CON1' results first and discuss these before moving onto the presentation of the sensitivity experiments. These also might be best presented as groups of experiments with the results and discussion presented together for each group. It may just be personal preference, but I would find it easier to be able to look at the temperature, precipitation and pressure differences together for one experiment (or set of related experiments) before going on to consider the next experiment. If a restructure is undertaken, I would move the comment about Antarctic temperatures (p6, line 17 and line 22-23) into the results/discussion of 'CON1-FREE', e.g. 'Initial tests showed ....', 'This was resolved by ....' giving 'CON1-FREE' results as shown in Figure ...

Response: The authors agree with the reviewer that the paper could be structured in another way; however, having the temperature, mean sea level pressure and precipitation plots all in the same place allows easy comparison between the control runs and each of the perturbed runs within the same panel. Furthermore, the current structure provides an overview of all the main features of each simulation and then, based on those interesting features, goes on to explain them. A reader can then either quickly browse through the overall results from the experiments (i.e. see how the control and perturbed simulations compare against each other) or look in more depth at the more

speculative scientific interpretation as to why we see the results presented in section 3. Therefore we would like to keep the current format as it is.

Technical comments p3, line 7: list Bi et al (2013) before Frauen et al. (2014) and perhaps note that the Bi et al. paper it is the ACCESS1.0 version that is most relevant. Response: Changed order and included "(primarily ACCESS1.0)" in the sentence.

p3, line 5: suggest adding 'configured similarly to ' before 'Hadley Centre ...' Response: Changed as suggested.

p3, line 9: delete repeated 'the' Response: Corrected.

p3, line 19: 'constraint' mis-spelled Response: Corrected.

p5, line 8: delete 'the' before 'ATMOS_PHYSICS2' Response: Deleted.

p6, line 5: Might be worth noting that the grid-cell values shown are the mean across the tiles in the grid-cell, assuming that is the case. Response: At the end of the sentence referred to we have included, "... (values are the grid-box mean across all surface tiles)."

p7, line 4: 'The first three experiments ...' not four. Response: Corrected.

p7, line 9: insert 'is the' before 'same' Response: Included.

p8, line 29 and 30: western Pacific, western Indian Ocean? Response: Yes, corrected as suggested.

p9, line 6: south-east of the Amazon? Response: Yes, corrected as suggested.

p9, line 10: the remote responses in the AUS10K temperature show some similarity to the con2-con1 differences. Do you think this is just coincidence? Response: Given that the changes in CON2-CON1 are not statistically significant then it is likely to be coincidence.

p9, line 12: east of the continent? Response: Yes, corrected as suggested.

p9, line 32: did you mean south-east, as this would be more consistent with the temperature anomaly? Response: Yes, corrected as suggested.

p10, line 6: 'increased precipitation coincides' Response: Corrected as suggested.

p10, line 15: add 'be' before 'representative' Response: We have changed the order of that sentence to read ('refs' refer to the existing references already there, which are unchanged): "Accepting that ACCESS (Ackerley et al. 2014; 2015) and other GCMs (Yang and Slingo, 2001; Dai and Trenberth, 2004; Dai, 2006; Dirnmeyer et al., 2012) produce convective rainfall too early in the day relative to observations, the same..."

p10, line 17: 'assess' mis-spelled Response: Corrected.

p12, line 6: add 'Antarctic' after 'allowing the' Response: Changed as suggested.

p12, line 26: delete space between T and 1.5 Response: Corrected.

p12, line 27: delete 'the' at start of line Response: Deleted.

p16, line 10: 'an' not 'and' towards end of line Response: Corrected

p16, line 13: Should be figures 11(a) and 11(e) not (b) and (f) Response: Corrected.

p16, line 31: replace 'or' with 'of' Response: This part of section 4.2.4 has been changed considerably and this suggestion no longer applies.

p17, line 5: ')' after 'respectively' Response: This part of section 4.2.4 has been changed considerably and this suggestion no longer applies.

p18, line 1: 'the' before 'imposed' Response: Corrected.

p20, line 3: 'GCM' instead of 'GCMs' Response: Corrected.

p20, line 21: 'model' not 'mode' Response: Corrected.

p21, line 4: 'of' repeated Response: Corrected.

p22, line 13: 'Cook' needs capital Response: Corrected.

Figure 1, middle column: the orange line is not defined in the figure caption (though the three-hourly input is mentioned for the right column but not shown?). Is the thickness of the black line significant or just for readability? Response: The authors have adjusted the figure in question so that the full caption can now be seen. The thickness of the black line is just for readability.

Figure 11: The colour bars in panels (a), (b), (e), (f) appear to be swapped. Response: The colour bars have been corrected.  
* * *
[Figure]

[Figure]

**Fig. 1.** The difference in (a) surface long-wave emission (upwards, W m -2), (b) upwards latent heat flux and (c) upwards sensible heat flux between CON1 and FREE.

[Figure]

---

## Author Comment (AC2) · 21 Apr 2016

Reviewer general comments: In this paper, the authors argue that the impacts of land temperature anomalies on the atmosphere can be investigated by imposing constraints on an atmospheric GCM, in a similar way to simulations with sea surface temperature anomalies. A method of constraining land surface temperatures in the ACCESS model is presented and it is shown that the simulated climate, including the diurnal cycle, matches well the unconstrained result. A set of experiments, partly motivated by earlier studies, with land temperature in various regions changed by 10 K is then presented. These are of considerable interest and do provide a 'proof of concept'. The Discussion section then presents further results that explore physical mechanisms and make rather lengthy comparisons with C1 GMDD Interactive comment Full screen / Esc Printer-friendly version Discussion paper other studies. The final section makes

some conclusions that seem overstated, and includes consideration of possible further experiments in unnecessary detail. In some respects, these sections go beyond the initial aim of the paper. Much of the presentation is very good and the work is potentially an excellent contribution. However, various limitations, noted below, also indicate a need for a considerable revision. Some reduction in the text could be needed, but some of the material might be better considered in a further paper in a different journal.

Authors' response: The authors would like to thank the reviewer (Dr. Ian Watterson) for his insightful, constructive and supportive review of our work. We have endeavoured to respond in detail to the comments raised and hope that we have answered those issues sufficiently.

Specific comments

Reviewer comment A: Prescribing land surface temperatures within a GCM could be a fairly simple exercise. In the case of ACCESS (2.2.1), the specification of surface temperature is evidently complicated, and the description given may not be well understood by a reader not familiar with the MOSES scheme. Eq 2 does not readily follow from Eq 1. It is not clear how 'surface' temperature relates to that of the first soil layer (of depth 0.1m), what $G0$ is and how it relates to $T^*$ and $Ts$. How does step 'n' relate to the final, etc?

Response: The authors accept that there appears to be a large step between Eq. 1 and 2; however, the intention was not to present the equations as such. Eq. 1 was presented just to illustrate the scheme used in the explicit calculation and Eq. 2 was intended to show where we have actually changed the code. Given that we state in the paper that we "only describe the equations that are changed... to prescribe $T^*$" we have therefore removed Eq. 1 as it is not changed in the new version of the code. We have adjusted the paragraph as follows (new paragraph following new Eq. 1, which was the old Eq. 2):

"Where $Ts$ is the temperature of the first soil layer beneath the surface at the end of the

previous time step (K), Rs is the net radiation (SW and LW) into the soil layer through the surface (W m-2), A* is the coefficient to calculate the surface heat flux (W m-2 K-1), Cc is the areal heat capacity of the surface (J m-2 K-1), Dt is the time step length (s), Tprev* is the surface temperature from the previous time step (K), all other variables have the same definition as described above. The term Cc/dt(Tprev* - Ts) represents the conductive energy flux from the first soil layer to the surface of the soil during the previous time step and is equivalent to the ground heat flux (G). More details on the derivation of Equ. (1) can be found in Essery et al. (2004) and Best et al. (2005)."

Given that we have only indicated where the code has been changed and cited all of the necessary literature, it would be easy for another person to find these equations within the model and re-produce our results. Furthermore, as we have not changed any of the other code in the MOSES scheme it is unnecessary to include the full derivation of Eq. 1 (what was Eq. 2 in the first review) when it is available in the cited literature. Also, in order to provide more information for readers we have also included an extra references to Kowalczyk et al. (2016), Best et al. (2005) and Essery et al. (2003), which have more details on the surface schemes employed by ACCESS should a reader require more information.

Reviewer comment B. Related to the specification of surface temperature anomalies should be a consideration of the energy fluxes associated with it. From P8L1 on, terms 'heating' and 'cooling' are used without explanation. Are these are the implied fluxes needed to keep a surface layer at the prescribed temperature? In any case, the surface (anomaly) must be then heating or cooling the air, which is clearly important. In fact, a warm surface might appear to be losing heat -so cooling, in that sense. Further description of these processes is needed. Response: We agree fully with the reviewer that the use of 'cooler' and 'warmer' (and similar language) is incorrect and we have removed such wording from the text and replaced it with e.g. increased / decreased land surface temperature. As for the surface temperature specification and the fluxes—we do not alter the fluxes and only change the surface temperature. The fluxes are allowed

to respond to the surface temperature perturbation. Using the 1.5 m air temperature therefore provides an indication of whether increasing or decreasing the land surface temperature is having the desired impact on the atmosphere above it. Furthermore, given that all of the atmospheric responses are consistent with the imposed surface temperature perturbations (i.e. increased convection over the Amazon when surface temperatures are increased), this implies that the surface fluxes must be responding in a sensible way to the perturbation.

Reviewer comment C. The presentation of COM2 results (from P9L23 on) seems excessive. If the initial condition change is merely a tweak in the atmosphere, then one would expect no impact on the climate. Indeed there seems to be no statistically significant differences, so what is the interest in the CON2-CON1 results? (Presumably, they do indicate a typical pattern of random or weather-induced differences in 50y means.) A case could be made for averaging the two and using this as the base for other results. If not, some amendment and reduction in the presentation can be made.

Response: Again, we agree with the reviewer that we should expect no impact on the climate, which is exactly why we have included that analysis. It provides assurance to a reader in that they will be able to re-produce our results without needing the same initial conditions. It is possible (given such a significant change to the surface scheme) that starting the model from a different initial condition may result in an unforseen drift in the mean climate state. The CON2 experiment simply shows that a user does not need the same starting conditions in order to run the simulation, which we think is an important (and reassuring) result to show. It is also appropriate to draw brief attention to CON2-CON1 in this case is given the journal (GMD).

Reviewer comment D. Regarding the tropically forced wave-like patterns (P14), while westerlies will aid propagation, other studies have shown that non-zonal components of a background state can also aid propagation through easterlies, especially into the winter hemisphere (which seems favoured in 9 b, c and f). Early studies include Schneider and Watterson (1984, J Atmos Sci) and Watterson and Schneider (1987, QJRMS),

and these are built on more recently by studies such as Zhao et. al (2015, J. Climate). Could more recent studies than the three in 4.2.2 also be considered?

Response: The cited articles (Ambrizzi, Hoskins, Karoly, Jin etc.) provide very close examples of the processes that appear in the simulations described in this paper, which is why they were chosen (despite them being pre-year 2000). It is clear from Fig. 9b that there is wave activity in both hemispheres, which corresponds with the surface temperature perturbation extending beyond the region of easterly flow. In Fig. 9c there is virtually no stationary wave activity in the summer hemisphere (NH) and only wave activity in the SH, again, where the surface temperature perturbation extends into the mean westerly flow. Finally, in Fig. 9f, the same process occurs, namely that the southern end of the surface temperature perturbation extends into the SH westerlies (and waves can be seen in the SH) whereas the northern limit is embedded well equatorward of the mean NH westerly flow. Therefore, in the cases indicated by the reviewer, all of them are consistent with the easterlies acting as a barrier to Rossby wave propagation in the climatological mean. The authors therefore stand by the presented interpretation. Nevertheless, the reviewer raises a very important point that is not considered or even discussed in our work (i.e. that the u=0 critical latitude assumption is not always appropriate). The experiments here could easily be applied to run experiments akin to Schneider and Watterson (1984), Watterson and Schneider (1987) and Zhao et al. (2015) and therefore acknowledgement of these papers must be included. We have included the following paragraph at the end of Section 4.2.2 to address this:

"Overall, the circulation responses to both of these tropical surface temperature perturbations are consistent with the results of Hoskins and Karoly (1981), Hoskins and Ambrizzi (1993) and Jin and Hoskins (1995). Nevertheless, there are cases where the cross-equatorial meridional flow can allow Rossby wave propagation through easterly flow (as discussed in Schneider and Watterson, 1984; Watterson and Schneider, 1987; Zhao et al., 2015). For example, Zhao et al. (2015) show that wave sources in

the summer hemisphere can excite wave activity in the winter hemisphere if the merid-
ional flow is from the summer to the winter hemisphere. Therefore, the idealised GCM
with prescribed land surface temperatures in this study is likely to be useful for running
similar experiments that address all of these features (where easterlies do and do not
act as a barrier to wave propagation)."

Reviewer comment E. A potentially important result of the pair of AM experiments (P16)
is that despite the large amplitudes (+10K, -10K) the response seems apparently linear,
differing only in sign. Could this be highlighted? In any case, some of the discussion
and comparison with earlier studies seems rather speculative. Does convection really
act similarly to topography (P16L25)? Indeed, is there an explicit parameterization of
the effect in the model? If not, what is the mechanism?

Response: The authors agree that it is worth noting the linear response and we have
included the following in the first paragraph: "The atmospheric responses to the $\pm 10$
K surface temperature perturbations over North America also appear to be of almost
equal and opposing sign in each respective simulation, which suggests the circulation
and precipitation respond in a linear way to the different surface temperature condi-
tions."

With respect to the drag caused by convection—one of the other reviewers brought
new literature to light that helps to elucidate the processes at work as a result of the
AM10K and AMm10K experiments (different from those stated). We have removed the
last two paragraphs in Section 4.2.4 and replaced them with a new discussion (and we
have also updated Figs 11d and h). The new literature provides a basis for a different
set of fascinating experiments that could be run with this new version of ACCESS to
investigate the impact of continental heating on the sub-tropical high-pressure cells.
The new literature is therefore more appropriate to discuss the AM10K and AMm10K
experiments (and not the original discussion).

Reviewer comment F. Despite rather extended discussions (section 4), the comparisons of the perturbed temperature cases with earlier studies can only be qualitative –the resulting temperature anomalies are different. The conclusions (P17L15) 'clearly show.. agree with previous studies' are rather over stated. Even at P1L12,'seems qualitatively consistent' might be enough. This links to the aims of the paper, as noted above.

Response: The authors accept that there is speculation attached to the discussion section; however, we feel that such speculation is warranted in order to showcase the gaps in the literature where this model may be useful for increasing scientific under- standing of atmospheric processes. The aim was not just to produce a new version of the model, but to provide several examples (and suggestions) of how it can (or could) be used. We feel that such speculation may encourage others to use this new model setup over a broad range of applications. Nevertheless, the language used is inap- propriate given that the results are mainly a qualitative comparison of known physical processes. We have therefore taken the reviewer's advice and changed some of the language in Section 4 (such as "clearly shows" to "is consistent with") to reflect this.

Minor comments P1L19 Land temperatures also respond to the simulated weather, of course. Response: We accept this but it does not change the point of the sen- tence, which is that land temperatures are not normally prescribed and so can respond (through the atmosphere) to the prescribed SSTs.

P3L4 Since Bi describes two (coupled) versions, the one most like the model used here could be identified (presumably ACCESS1.0, as used in CMIP5, but at reduced resolution here). Response: We have adjusted the text to refer to this setup being more akin to ACCESS1.0.

P3L7 It might be more usual to state that 'Physical processes represented in the model include'. There are explicit components, in addition to some parameters. Response: Changed as suggested.

P3L9 'the the'. Response: corrected.

P3L17 Does 'all' include FREE? Response: Yes, it does include FREE. This is done in order to be sure that the FREE soil moisture is consistent with those in the prescribed experiments.

P3L17. Does 'deep soil' mean layers 2, 3, 4? Is there flux through the bottom of 4? Response: yes, this does mean layers 2, 3 and 4 and there is zero flux boundary condition at the bottom to ensure energy conservation. The text has been updated to state, "... and deep soil temperatures (i.e. on all four levels described above)...".

P4L1 Should this be 'SF_EXCH' –as in the Figure? Response: Yes, corrected.

Fig 2 It seems the ' ..hourly interpolated temperature field' is in the middle column. The detail in the third column is not visible and seems to create an unwieldy file. It might be simplified. Response: The figure has been simplified and the missing text is now visible.

P6L1,3 Does 00:00:30 mean 30 seconds after midnight? Should the first 00: be dropped? Response: Yes, corrected as suggested.

P6 L17, 23 (and elsewhere) 'reduce' is being used in an uncommon, intransitive way. Response: We have considered the use of 'reduce' and have altered the manuscript to account for this (i.e. remove and replace).

P6L27 'PRES', but (6) has lower case. Response: Corrected.

Table 1. 'Maritime Continents'? Response: Corrected.

P7L8 'at the'? Response: Corrected to "as the".

P7L9 The soil temperatures and moisture are also prescribed, it seems. Response: We have included "... uses prescribed, climatological soil moisture, deep soil temperatures, SSTs..." for the FREE description in Section 2.3 to reflect what is said in Section 2.1.

P8L4 One might doubt if the processes in the response to such large (10K) anomalies

can be known, from observations. Is this magnitude chosen to improve statistical significance of responses, given some expectation of linearity? Response: The value of 10 K was chosen to maximise the atmospheric response to the imposed temperature changes (i.e. so they would be obvious). Given that the atmospheric responses compare well (qualitatively) with those within the literature we are confident that our model is doing what we expect (even though the perturbations are exceptionally large, the atmospheric response is physically realistic). Ultimately, the experiments show how the model we have developed could be used in the wider scientific community.

Fig. 3 Would grid square shading, as in Fig. 4, give a clearer depiction than the interpolated lines? Some explanation of the different usage could be added. Response: The difference is primarily due to aesthetics and does not require explanation in the text. We have looked at using grid-square shading for temperature and the figures appear less 'busy'; however, the current figures show interesting features more clearly (Fig. 3), such as the stationary Rossby waves, and so we are going to leave Fig. 3 unchanged. The precipitation plot, given the nonlinear colour bar, works better as grid box shading.

P8L22 (and later) If the 1.5m temperature is an interpolation from the surface (subject to parameterisations) then it will be strongly constrained to the prescribed land and sea values. Temperature at the first atmospheric level would be a stronger indication of an atmospheric response. A brief comment justifying the focus on 1.5m seems warranted. Response: The 1.5 m temperature provides the 'first check' on whether our setup is correct, as it is so close to the surface. If the 1.5 temperatures do not respond as expected then we know instantly that we have done something wrong. Furthermore, given that the atmospheric responses to each of the surface temperature perturbations are also what we expect (e.g. increased convection over the Amazon), evaluating the temperatures on the first level of the atmosphere would not add more insight beyond what is already presented.

P9L5,7 'alternating' does not seem a good description here –although it is better for precip. Response: The anomalies do alternate between positive and negative (and are

clearer than the precipitation ones) so we would prefer to keep that description.

P9L10 lower case'k' Response: Corrected.

P10L16 'be representative' Response: Corrected as suggested.

P11L11 'Similarly' is odd. As before, CON2-1 is expected to be the same, but CON1-FREE is the main test. Response: Agreed, but this is important in the context of the new model developed here. It re-iterates that the initial conditions (as expected and desired) are unimportant.

P11L13 Presumably MSLP is an extrapolation from a surface that is now warmer, so one might expect it to be lower, even if the surface pressure is unchanged. How much of the lowering might be due to this? Is the surface pressure different? Response: We accept that the surface pressure may be unchanged despite a lowering in MSLP; however, given the circulation response shown in Fig 8 (i.e. ascent primarily over the land and descent over the ocean) it would seem likely that the surface pressure is also reducing due to the higher land surface temperatures in the ALL10K simulation. The MSLP field, in this case (Section 3), provides a simple illustration of the impacts of the surface temperature perturbation experiments and are not key to the interpretation. The main interpretation is in Section 4 (and Fig 8) where the circulation changes are shown.

P12L25 It seems the mean surface temperature is the same, but there tends to be more snow in CON1. How does that influence T1.5? Response: We state that, ". . . snow melt is prevented earlier in CON1 than FREE and so snow amount are, on average, higher in CON1 during the cold season, which causes T1.5 to be systematically lower", which should answer this point.

P13 Consistent with the earlier suggestion regarding CON1 and 2, this 4.1.2 seems unnecessary. Response: Given the brevity of this section and that this is a model development paper, we think this section should remain to show any potential future

users of this model that the initial conditions they use do not matter. Again, while this is expected, it should be shown for completeness.

P13L9 Are the SSTs unchanged in Chadwick's warmer-land run? Response: Yes. They only change the solar constant and use prescribed SSTs.

P13L22. Would vertical velocity closer to the centre of the moisture column (e.g. 850 or 700hPa) be an even better match? Response: We would expect areas with increased deep convection to extend through 500 hPa, which is why this field was chosen to compare with the precipitation. Furthermore, given the very high pattern correlation (-0.69), it appears that the 500 hPa omega field is useful for explaining the changes in precipitation.

P14L2 'increases subsidence over India' is not clear. Response: Changed to, "... results in positive differences in $\omega 500$ for ALL10K relative to CON1 over southern India, which would suppress precipitation."

P14L7 Often Rossby waves are excited by the latent heat from rain formation. Does this provide the 'imposed heat sources' that are described here? If so, does the reduced rainfall over the seas in MC10K counter the effect of enhanced rain over the land? Response: Yes, the increased convection over the Maritime Continent is from the increased surface temperatures (i.e. imposed heat sources). The authors see how this language is a bit vague so we have been more explicit and changed the text to, "... propagating away from the imposed tropical heating sources (Gill, 1980), which in this case are from increasing surface temperatures by 10 K and the resulting increase in latent heat release (inferred from the increase in precipitation, see Figs. 4(d) and (e))." There does not seem to be any reason why the reduced rainfall over the sea would counter the effect of enhanced rainfall over the land. Given that there is an increase in ascent over the islands (as indicated by the increased rainfall) from increasing the surface temperatures, then there should be subsidence in the region surrounding that ascent (suppressing rainfall). Such an impact can also be seen in the AMA10K simulation, and to a large extent in the ALL10K simulation, where ascent is enhanced over the land with subsidence over the ocean.

P16L10 'and increase' Fig 11 labels are bulky –with m10K partly missing. The bars are incorrect (swapped) in a, b, e, f. Perhaps simplify, with bars combined for the pairs? Response: Changed the text as suggested and corrected the colour bars. The figure titles are a bit bulky but we feel they are necessary to make it easier to know what each panel is showing without having to re-read the caption.

P17L21What supports 'the local response is governed by the strengthening .. of existing circulations'? Response: We agree that this statement is vague and therefore unnecessary. We have removed it.

———————————————

---

## Author Comment (AC3) · 21 Apr 2016

Reviewer general comments: Overall this is a well-written paper introducing a very novel experimental design: prescribed land-surface temperature AGCM experiments. Some interesting but idealized experiments are also introduced to demonstrate that the approach gives reasonable responses. I recommend publication subject to some minor revisions, listed below.

Authors' response: The authors would like to thank the reviewer (Prof. Noel Keenlyside) for his insightful, constructive and supportive review of our work. We have endeavoured to respond in detail to the comments raised and hope that we have answered those issues sufficiently.

Main concerns (1) The description of the response to the NH heating, which seems not

the most relevant. The study from Miyasaka and Nakamura (2005) is more relevant.

Response: Having read Miyasaka and Nakamura (2005), the authors fully agree with this issue. We have removed the original description and adjusted Figure 11 accordingly. We have also included the following paragraphs to explain the process: "Miyasaka and Nakamura (2005) show that the land-sea thermal contrast along the west coast of North America is important in causing the formation and maintenance of the Northern Hemisphere, summertime sub-tropical high pressure cell over the North Pacific. Miyasaka and Nakamura (2005) show that the increase in low-level potential temperatures from boreal spring to summer over the North American continent in July (and May) acts to increase cyclonic vorticity (cyclone stretching) over the continent, which strengthens the northerly flow along the west coast. Strengthening of the northerlies then increases the advection of polar air over the ocean, enhances evaporation from the ocean surface and encourages the development marine stratocumulus, which all act to reduce SSTs. The cooling of the air column causes subsidence (visible at 500 hPa, see Fig 8(d) in Miyasaka and Nakamura, 2005) and enhances the anticyclonic circulation (vortex compression) within the sub-tropical high-pressure cell over the ocean and strengthens the northerly flow and subsidence further.

Interestingly, the differences in circulation in Fig 11(c) are qualitatively very similar to those produced by Miyasaka and Nakamura (2005), which suggests that increasing North American surface temperatures by 10 K may result in a strengthening of the Pacific sub-tropical high pressure cell. To illustrate this further, the values of $\omega 500$ from CON1 (black solid and dashed lines) and the difference between AM10K and CON1 (coloured shading) are plotted in Fig. 11(d). The largest increases in subsidence (red shading) at 500 hPa occur over the centre and to the north of the maximum subsidence in CON1 (Fig. 11(d)), which may indicate a strengthening and northward shift of the summertime high-pressure cell. Conversely, the opposite circulation anomalies occur in the AMm10K simulation (and with very similar magnitude), which suggests that the same process may be reversed by decreasing North American land surface temperatures (also seen in in the $\omega$500 field, Fig. 11(h)). It is therefore likely that increasing or decreasing the North American land surface temperatures in ACCESS may act to enhance or weaken the strength of the Pacific sub-tropical high pressure cell (given that SSTs in AM10K do not respond to and feedback on the atmospheric circulation in the way described in Miyasaka and Nakamura, 2005). These results therefore indicate that this version of ACCESS (with prescribed land surface temperatures) may be useful for investigating the impact of regional land-sea thermal contrasts on the location and strength of the summertime sub-tropical high pressure cells, for example."

This new description (based on the suggested literature) really showcases a potentially fascinating application of this new version of ACCESS and fits much better with the results presented.

(2) I am not convinced about that there is a statistically significant response over the SH westerlies induced by Australia heating.

Response: We have now plotted the places where the changes in the winds are statistically significant (Fig. 10), with much of the annual mean change significant. Interestingly, the strongest significance seems to be in DJF with little significant change in JJA. Further discussion on this is given below when the reviewer raises those points. Nevertheless, we feel that the line discussing the change in the SH Hadley Cell is too speculative as we do not show any evidence for this (or any other reasoning) and so we have deleted it from the text.

Minor points Pg2, L15, Without having read the entire paper, I find aim 2 a little hard to follow because you do not say that you prescribe the very same land surface temperature from the freely varying run, and that this implies that the experimental design does not introduce spurious effects.

Response: We have changed aim 2 to be: "Show that simulations with prescribed and freely varying land surface temperatures (with the land temperatures in the prescribed run being derived from the freely varying simulation in order to avoid spurious effects)

[Figure]

are climatologically comparable."

Pg3, L15-20, I would have imagined that soil moisture would be a key variable to pre-
scribe to the atmospheric model to capture the surface energy budget. I wonder what
are the implications of fixing it to climatology in the 10K experiments? I think you should
at least acknowledge that this might impact the results of the surface heating experi-
ments. It might be worth mentioning here that snow cover is simulated? I wonder if you
were to prescribe it, whether you would fix the deviations of CON from FREE.

Response: The authors agree with the reviewer but in this instance we decided to pre-
scribe the soil moisture in order to constrain the surface as much as possible. This was
to prevent the soil moisture from responding to the imposed temperature perturbations,
which could have induced extra feedbacks. Nevertheless, removing such a constraint
would be a sensible development of these simulations and also would be easy for any
other users of the code to undertake. We have actually repeated the AMA10K ex-
periments with and without the soil moisture constraint as part of our next phase of
development and we have included the figures below. FIG 1: (a) Difference in MSLP
(hPa) between the AMA10K and CON1 experiment and (b) for AMA10K – CON1 with
freely evolving soil moisture. FIG 2: (a) Difference in precipitation (%) between the
AMA10K and CON1 experiment and (b) for AMA10K – CON1 with freely evolving soil
moisture. In both experiments there is a reduction in MSLP over Amazonia; however,
the stationary wave-like response in FIG 1(a) is non-existent in FIG 1(b). The reason
for this is clear in FIG 2 as when we prescribe soil moisture in (a) we get increased
precipitation (i.e. increased convection, which can then cause the wave formation)
whereas in (b) the precipitation is lower and suggests a weakening of deep convection
(i.e. less impact on the large-scale circulation). It is likely that the moisture in the soil
is evaporated away and therefore the local moisture source for rainfall in the Amazon
is reduced. The reduction in MSLP is likely to be caused by increased dry convection
(i.e. a dry heat-low) from the surface heating once the moisture has evaporated. If we
extrapolate out globally, then it is likely that the circulation response in the ALL10K simulation would also be weaker than presented in our work here if the land-based tropical convective centres are sensitive to the local soil moisture content. We have included the following in the future work list as it acknowledges the impact of the soil moisture constraint: "..., which could have an impact on the modelled climate. For example the global circulation response in the ALL10K experiment may not be as strong once the local moisture supply from the land-based convection has been evaporated away." With respect to the snow cover—we have included "...(and snow cover is not prescribed)" in Section 2.1 to point out that we do not prescribe it. Prescribing the snow remains an area of future development and would be useful for other experiments but is beyond the scope of this paper.

Pg3, L30, In my version latent heat is labelled here and in the equation as labdaE, while in the figure 1 it is LE.

Response: Changed in Fig. 1 to be $\lambda$E.

Pg 8, s30, I am surprised that T1.5 does not heat further. It seems rather artificial that up to 8K temperature gradient can be formed in the lower 1.5 m of the BL. Some discussion is required of how this can be possible.

Response: Looking at the surface energy balance equation, heat may be lost (or gained) from the surface through long-wave emission, conduction/convection into the air (sensible heating), evaporation and conduction into the ground (ground heat flux). If surface temperatures are increased, and all four methods for re-distributing that heat are equally important, then only one quarter of the extra energy will be available for sensible heating above the land surface to increase T1.5. The long-wave emission will distributed over the whole atmospheric column, latent heating does not increase the air temperature until that energy is re-released in condensation (above the surface) and the ground heat flux is directed into the soil. Furthermore, the values of T1.5 are calculated by interpolating between the surface and the lowest model level. Therefore, the T1.5 response also depends on the global atmospheric response to the increased land

surface temperatures, which will be relatively weak as we are only warming $\sim$33% of the global surface (the other 67% of the globe has unchanged, prescribed sea surface temperatures).

Pg 9, s20, Is there any reason to expect changes in the initial conditions should lead to a significant difference on these timescales?

Response: There is no reason why we would expect there to be a difference but we think it is important to show this so that future users of this model know that there is no impact on their simulations from not using the same starting field that we used. There is a chance that changing the initial conditions could lead to drift in the simulations (for example, the build up of snow in CON1 relative to FREE could have been sensitive to the initial conditions). Given that no such drift occurs in CON2 and the scope of the journal (model development), the authors feel that it is worth confirming this point.

Pg 10, s15, "including ACCESS" is misplaced.

Response: The text has been rearranged.

Pg13, s5, Again, I am not clear why you would expect a difference between the CON1 and CON2 simulations. Memory of the atmospheric initial conditions is lost very quickly, and should be gone within a several months I think you should make this clear.

Response: We do not expect a difference and it is to confirm that nothing unexpected happens when the initial conditions are changed. To make this clearer in this context we have adjusted the end of the second sentence of 4.1.2 to read:

"... shows that this model setup is reliable for other users to perform idealised simulations without the need to use the same initial conditions as this study."

Pg 15, s5, while the arguments given seem reasonable, it seems hard to discount completely the extent of diabatic heating, which is surely greater the AMA case (as seen in the precipitation field). I think you should be clear about this. Are the responses more comparable if they are scaled by the amount of diabatic heating?

Response: The reviewer makes a good point here and this needs to be accounted for in the text. We have included the following sentence to cover this at the end of the paragraph referred to: "Nevertheless, the larger areal extent of the diabatic heating (and higher precipitation amounts) in AMA10K relative to MC10K is also likely to be an important factor in the different wave responses between those two simulations." Given this is a proof of concept paper and that the explanation given qualitatively ties in well with the results in the cited literature, a more in-depth look at the size of the diabatic heating region is unnecessary in this case. Nonetheless, it is something that a future user could look at in more detail given these initial sensitivity studies.

Pg15, s30, The SH Hadley Cell should be present during JJA (i.e., strongest in the winter hemisphere). Are the changes in the SH winds statistically significant?

Response: The changes in the winds are significant in DJF but not JJA. We have therefore removed the line that included reference to the Hadley Cell circulation changes as it is speculative at best.

Pg 16, s5. It would be useful to put the anomaly surface heating into perspective. For example, could you please discuss it in terms of changes expected by the end of century? It would put the simulated responses into perspective.

Response: Looking at Figure 12.11 from IPCC AR5, Ch. 12, the surface air temperature responses we see over land in our experiments are comparable with those of the ensemble mean, end of century (2081-2100) land surface air temperature changes under RCP8.5. We have included the following at the end of Section 3.1 (T1.5 results), as it seems this is the most appropriate place for it: "Interestingly, in the experiments with higher land surface temperatures (ALL10K, AMA10K, MC10K, AUS10K and AM10K), the T1.5 responses are similar to those of the CMIP5 multi-model ensemble average for the end of the 21st century (2081-2100) under RCP8.5 (high greenhouse gas concentrations; see Fig. 12.11 in Collins et al., 2013). Similarly, the negative T1.5 anomalies over North America in AMm10K are of a similar magnitude to those simulated over

land at the Last Glacial Maximum (see Fig. 2 in Harrison et al., 2014)."

S4.2.4, NA experiments. The mechanisms proposed by Brayshaw et al. (2009) are more relevant to the NH winter time circulation and the NA Storm track. I think the work of Miyasaka and Nakamura (2005) is much more relevant. Takafumi Miyasaka and Hisashi Nakamura, 2005: Structure and Formation Mechanisms of the Northern Hemisphere Summertime Subtropical Highs. J. Climate, 18, 5046–5065.

Response: We fully agree with the reviewer here and have made the necessary changes to the text (see the response to the first main concern).

In terms of the conclusions: (1) I think you should mention in the first bullet point "(excluding Antarctica)" , or something along those lines. Perhaps the reasons for this are not clear, and don't need to be explained as the experiments are still very interesting. Response: We have included "(excluding Antarctica)" as suggested.

(2) It is not clear to me that there really was a significant change in the SH circulation in response to Australian heating. Response: The change is significant (annual and DJF-mean), but our explanation was very speculative with no real evidence to back up the cause (i.e. Hadley circulation changes). We have therefore removed that sentence and left the rest of the discussion about the SAM, which is more relevant.

(3) Also the explanation for the response to NA heating does not seem appropriate (see comment above). Response: We have adjusted the text and taken account of the literature suggested by the reviewer (see above).
* * *
[Figure]

**(a) AMA$_{10K}$ - AMA**

**(b) AMA$_{10KF}$ - AMA$_F$**

MSLP difference (hPa)

**Fig. 1.** The difference in mean sea level pressure (MSLP, hPa) for (a) AMA10K – CON1 (prescribed soil moisture) and (b) AMA10K – control run (both with freely varying soil moisture). In both cases the land sur

[Figure]

**Fig. 2.** The difference in mean precipitation (%) for (a) AMA10K – CON1 (prescribed soil moisture) and (b) AMA10K – control run (both with freely varying soil moisture). In both cases the land surface temperat